ecology, health and disease and epidemiology, computational biology

SARS-CoV-2, non-pharmaceutical interventions, mathematical models, epidemics, reproduction number

**Authors for correspondence:**
Marissa L. Childs
e-mail: marissac@stanford.edu
Morgan P. Kain
e-mail: morganpkain@gmail.com
Mallory J. Harris
e-mail: mharris9@stanford.edu

†Denotes equal authorship.

# The impact of long-term non-pharmaceutical interventions on COVID-19 epidemic dynamics and control: the value and limitations of early models

Marissa L. Childs[1,†], Morgan P. Kain[2,3,†], Mallory J. Harris[2,†], Devin Kirk[2,6], Lisa Couper[2], Nicole Nova[2], Isabel Delwel[4], Jacob Ritchie[5], Alexander D. Becker[2] and Erin A. Mordecai[2]

[1]Emmett Interdisciplinary Program in Environment and Resources, [2]Department of Biology, [3]Natural Capital Project, Woods Institute for the Environment, [4]Department of Microbiology and Immunology, and [5]Department of Computer Science, Stanford University, Stanford, CA 94305, USA
[6]Department of Zoology, University of British Columbia, Vancouver, British Columbia, Canada V6T 1Z4

MLC, 0000-0002-8597-2161; MPK, 0000-0003-0605-7289; MJH, 0000-0001-8196-0430;
DK, 0000-0001-9588-1004; NN, 0000-0001-8585-1215; ID, 0000-0001-6671-3669;
EAM, 0000-0002-4402-5547

Mathematical models of epidemics are important tools for predicting epidemic dynamics and evaluating interventions. Yet, because early models are built on limited information, it is unclear how long they will accurately capture epidemic dynamics. Using a stochastic SEIR model of COVID-19 fitted to reported deaths, we estimated transmission parameters at different time points during the first wave of the epidemic (March–June, 2020) in Santa Clara County, California. Although our estimated basic reproduction number ($\mathcal{R}_0$) remained stable from early April to late June (with an overall median of 3.76), our estimated effective reproduction number ($\mathcal{R}_E$) varied from 0.18 to 1.02 in April before stabilizing at 0.64 on 27 May. Between 22 April and 27 May, our model accurately predicted dynamics through June; however, the model did not predict rising summer cases after shelter-in-place orders were relaxed in June, which, in early July, was reflected in cases but not yet in deaths. While models are critical for informing intervention policy early in an epidemic, their performance will be limited as epidemic dynamics evolve. This paper is one of the first to evaluate the accuracy of an early epidemiological compartment model over time to understand the value and limitations of models during unfolding epidemics.

## 1. Introduction

COVID-19, caused by the emerging virus SARS-CoV-2, rapidly expanded across the globe, overwhelmed healthcare systems, and has led to just under 4.0 million deaths with the pandemic still underway as of July 2021 [1]. It is just the latest and most widespread in a series of (re)emerging and expanding infectious disease outbreaks, including SARS-CoV in 2003, H1N1 influenza virus in 2009, Ebola virus in 2014 and Zika virus in 2016. Before effective vaccines and specific drug therapies are available at the start of emerging epidemics, non-pharmaceutical interventions such as social distancing, mask-wearing, diagnostic and serological testing, contact tracing, and quarantine are the best available tools to slow epidemics and to mitigate their health toll. Early in the COVID-19 epidemic, when epidemiological information was limited, governments and other decision-makers used models (e.g. [2–5]) to predict the spread of COVID-19 under various non-pharmaceutical interventions and to show the benefits of social distancing for reducing and delaying the

epidemic peak (i.e. flattening the curve) in an effort to prevent medical systems from becoming overwhelmed and to buy time for more effective treatments, testing capacity and potential vaccines to become available.

Early models can and should inform policy decisions in an epidemic as they are our primary tool for synthesizing early knowledge of transmission in order to define a plausible range of epidemic outcomes [6]. Such models also quantify trade-offs among proposed intervention scenarios to identify which responses will most efficiently slow exponential growth. During the 2014 Ebola epidemic, for example, early models promoted the use of contact tracing and sanitary burials to reduce transmission [7]. At the beginning of the COVID-19 pandemic, models identified the critical importance of social distancing to slow viral spread [2,8,9]. Early models also serve to illuminate difficult-to-observe processes and to test the implications of new information; for example, in February 2020, Hellwell et al. [3] showed that because of SARS-CoV-2 presymptomatic and asymptomatic transmission, contact tracing would be insufficient to curb the spread of the disease and strong social distancing would also be necessary.

Early models must be built rapidly and calibrated to data that are incomplete and of unknown quality, therefore it can be difficult to appropriately quantify uncertainty and to assess model accuracy in order to compare policy decisions. With ample time, alternative transmission scenarios can be thoughtfully compared and uncertainty well characterized [10]; however, the need for rapid decisions in the face of exponential epidemic growth makes such efforts infeasible for most early models. Though such real-time model assessment is rare, post-epidemic retrospective analyses of SARS [11–13], H1N1 influenza [14–21], Ebola [22–28] and Zika [29–33] have illustrated that much can be learned from emerging epidemics about the fundamental principles of disease transmission and epidemic modelling (e.g. the limits and utility of model complexity [15,23,32], the effects of population and geographic heterogeneity on disease dynamics [20,21,27], and the importance of stochastic models to capture uncertainty [13,25]). However, few of these analyses focus on models developed early in the epidemics. The accuracy and appropriate use of early models is rarely assessed, first because of a lack of emphasis and resources while the epidemic is underway, and later because early models tend to give way to more sophisticated and better-fitting models as more information and data are acquired (e.g. moving from using single transmission rate during the early COVID-19 lockdowns [34] to continuous-time human movement data [35]). Thus, we know less about the value and limitations of models that are built and applied early in an epidemic when epidemiological information is severely limited. Yet, this early model assessment remains vitally important because such models often inform policy and public opinion in real time, even if they are later revised. To understand and anticipate problems for future emerging infectious diseases, and to produce models that will be taken up by policymakers, it is critical to reflect upon the value and limitations of early models and to assess their accuracy over time.

The need for rapid model development with incomplete and uncertain data forces modellers to make a series of decisions and assumptions, many of which must be made with relatively little empirical evidence (e.g. about the proportion of infections that are asymptomatic and the transmission potential of asymptomatic infections). All early models of COVID-19 dynamics, for example, were constrained by

limited: (1) observations of unmitigated epidemic dynamics from which to inform key epidemiological parameters like $\mathcal{R}_0$; (2) information about the impact of preliminary interventions and (3) availability of testing, which made case data an unreliable indicator of epidemic magnitude and dynamics. Further, because of regional differences in socio-economic conditions and demography paired with difficult-to-observe case importations, disentangling local epidemic dynamics from policy interventions (and estimating the effectiveness of those interventions) proved difficult [36]. Data limitations also generate trade-offs between realistic complexity, parameter identifiability, computational feasibility and accuracy (see [37]), which require models to be designed around a targeted purpose rather than comprehensively describing all aspects of disease dynamics. Early in the COVID-19 pandemic, the need for rapid model deployment resulted in some researchers adopting a minimally complex statistical approach with the aim of producing near-term (e.g. one to five month) epidemic forecasts (e.g. [38,39]). However, because statistical models do not capture the underlying mechanistic transmission process, early COVID-19 statistical models were poorly suited to predict the effects of non-pharmaceutical interventions. On the opposite end of the spectrum, mechanistic, agent-based models sought to more precisely estimate the potential effects of different social distancing policies by incorporating population structures and individual movement (e.g. [2]). Such models were, however, computationally intensive, contained a large number of parameters, and did not always have publicly available code from the outset, making them infeasible to rapidly fit and simulate in many locations and relatively inaccessible to outside research groups and decision makers. Similar types of uncertainty and the need for rapid model development also led to the mixed success of early models of SARS [13], H1N1 pandemic influenza [14–17], Ebola [22–28] and Zika [32].

In addition to limited information, unreliable data and model trade-offs, a hurdle in the use of models early in the COVID-19 epidemic was the rapidity and heterogeneity of policy changes (especially in locations where interventions varied at a local level, for example in the USA). These changes, along with temporal variation in human behaviour (which often changed in advance of government interventions [40]) and extrinsic factors (e.g. seasonality), quickly rendered many models obsolete. Though many models were being continually updated (using, for example, more reliable case data due to expanded testing and cell phone-based mobility data), delays between changes in epidemic drivers and model improvements, their uptake, and public health decisions led to many models being used after the epidemiological environment was no longer reflected in the model's underlying assumptions. Furthermore, in the USA, for example, the fragmented COVID-19 response forced many state and local governments to rely upon a few highly publicized early models, which led to an outsized influence of some early non-peer-reviewed models (e.g. [38]).

Here, we quantitatively evaluate the successes and failures of an early epidemic model by retrospectively analysing an epidemiological compartment model that we developed in March 2020 for COVID-19 dynamics in Santa Clara County, California during the first wave of the US pandemic. While later development improved upon this model (see figure 1; [42]), here we use the early model as a snapshot in time, analysing it as an artefact rather than improving upon it

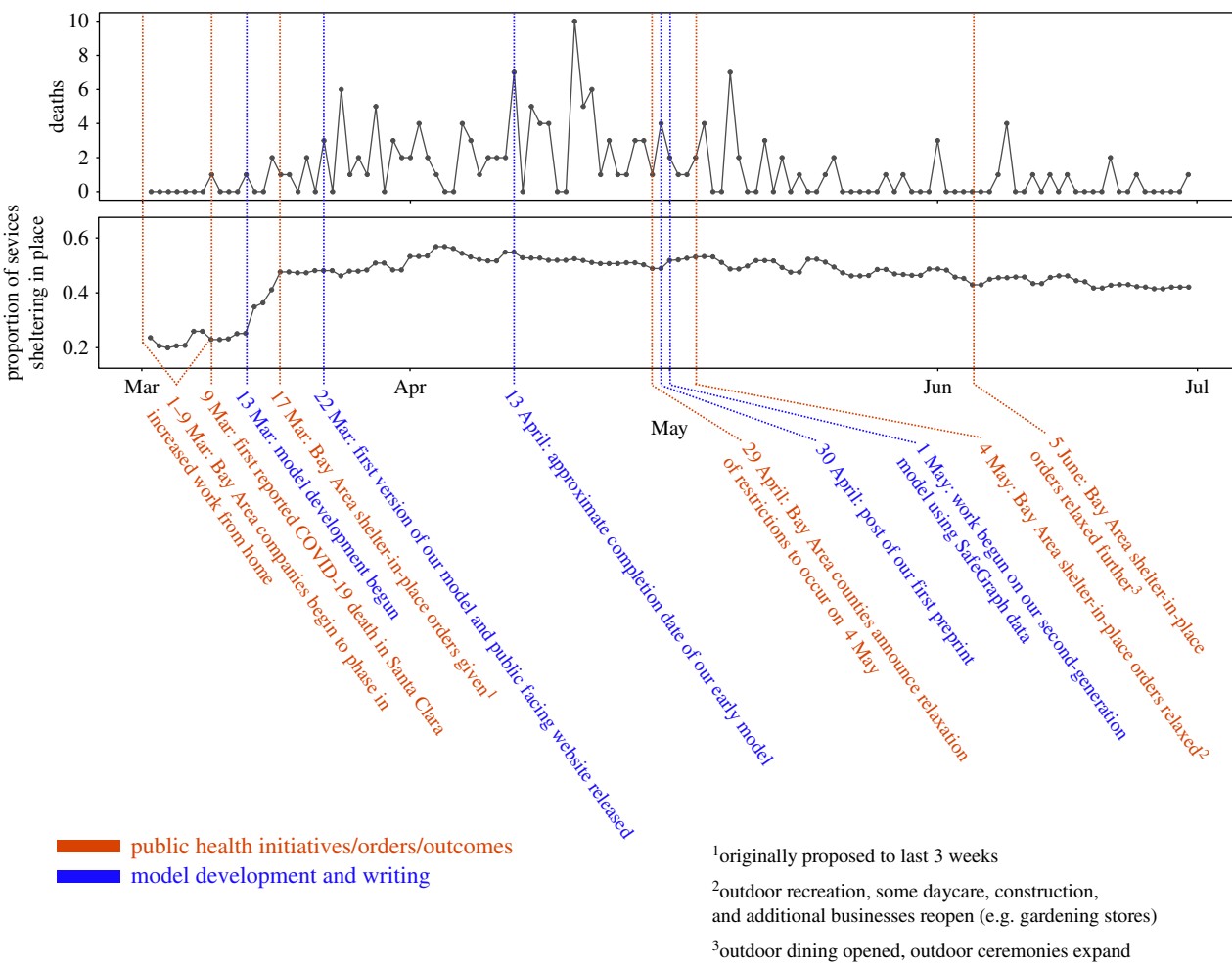

**Figure 1.** Timeline of the early COVID-19 epidemic period in Santa Clara County, California between February and May 2020. Reported COVID-19 death data (top) were used to estimate model parameters. The proportion of devices sheltering in place (bottom) [41] helps to illustrate how behaviour changed along the timeline of public health interventions and epidemic dynamics; these data were not used in our early model. (Online version in colour.)

[43]. That is, we intentionally seek to evaluate our model in light of the limited information and rapid decisions used to build it as well as the sparse data used to fit it. We critically gauge the strengths and limitations of this model by evaluating decisions that benefited and hindered model accuracy and potential model mis-specifications, and by quantifying the accuracy of predictions over time to understand why inaccuracy increased.

We began development on the model on 13 March 2020, 4 days after the first reported death in Santa Clara County and four days before the introduction of the San Francisco Bay Area shelter-in-place orders that applied to this county, which were initially proposed to last three weeks (figure 1). Our first aim was to deploy (within approximately two weeks) a public-facing user-friendly graphical model (http://covid-measures.stanford.edu/) that would allow users to adjust intervention parameters to help the public and local decision-makers understand the potential for resurgence if restrictions were lifted or relaxed too early and to evaluate viable exit strategies. Like other early COVID-19 epidemic models that were built for a specific purpose, we designed the model with the following considerations in mind. First, because we sought to compare the effects of various non-pharmaceutical interventions, we chose to build a mechanistic epidemiological compartment (susceptible–exposed–infectious–recovered: SEIR) model. Second, given early work

showing the potential for asymptomatic transmission [44] and our interest in symptomatic isolation as a potential intervention strategy, we broke up the infected classes into multiple compartments (asymptomatic, pre-symptomatic, mildly symptomatic and severely symptomatic). Because we were also interested in interventions triggered by hospital capacity thresholds, we included model compartments (state variables) for hospitalizations. Third, as we were interested in the implications of both initial, short-term interventions and longer-term exit strategies, we used a time-varying transmission parameter, $\beta_t$, to encapsulate the impact of non-pharmaceutical interventions on epidemic dynamics and control. Fourth, as case data were significantly biased at the time due to limits in testing capacity and access, we fit the model to local epidemic dynamics using daily reported COVID-19 deaths in Santa Clara County (which we assumed were more reliably reported than cases). We completed our first analyses on 13 April and posted a preprint on 30 April 2020 (figure 1). In the light of the narrow time window and these considerations, like many other early models we made a series of decisions (electronic supplementary material, table S1), some of which we deemed to be sub-optimal and improved in our later model, which we began developing on 1 May 2020 [42].

After developing the public-facing website, we originally used this model to estimate key epidemiological metrics and to evaluate the effectiveness of long-term intervention

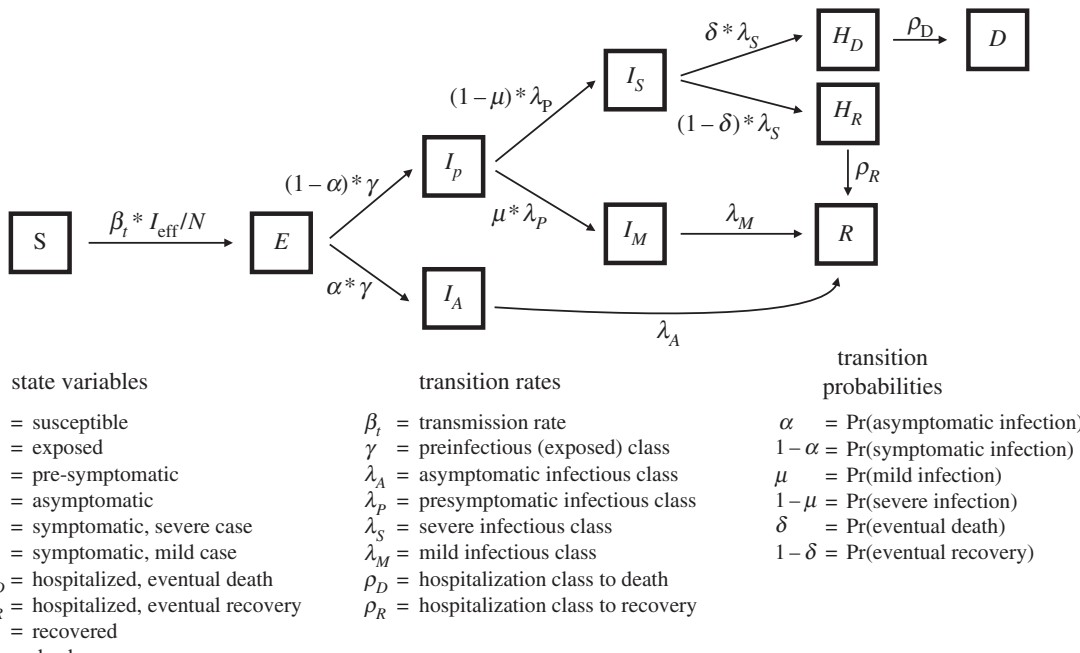

**Figure 2.** Epidemiological model structure. English letters in boxes designate state variables (model compartments), while the Greek letters $\beta$, $\gamma$, $\lambda$ and $\rho$ refer to transition rates between states; the Greek letters $\alpha$, $\mu$ and $\delta$ designate transition probabilities between states. We assume that the *per capita* transmission rate ($\beta_t$) is directly proportional to the effectiveness of social distancing ($\sigma_t$), such that $\beta_t = \beta_0 \cdot \sigma_t$, where $\beta_0$ is the transmission rate prior to any social distancing (i.e. $\sigma = 1$). The transition rate between the susceptible and exposed states is given by the force of infection ($\beta_t \cdot I_{eff}/N$), where $I_{eff}$ is the effective number infectious, which is equal to the weighted sum of infectious classes (in which the weights are the relative infectiousness of the infected classes: $I_{eff} = I_a\kappa_a + I_p{}^*\kappa_p + I_m\kappa_m + I_s\kappa_s$). See electronic supplementary material, tables S3 and S4 for details on the relative infectiousness ($\kappa$) parameters.

strategies, specifically focusing on Santa Clara County, California as a case study. We estimated the transmission rate under pre-intervention and shelter-in-place conditions, calculated reproduction numbers before and during interventions, explored the impact of long-term intervention strategies, and investigated counterfactuals to understand the impact of early intervention decisions. We now seek to answer the following retrospective questions: what did the model suggest about epidemic metrics, dynamics, and the impact of non-pharmaceutical interventions, and how did these estimates change over time? How accurately did the model predict epidemic dynamics going forward? For how long was the model accurate enough to be useful, and what limited its longer-term accuracy?

## 2. Methods

### (a) Model structure

We developed a stochastic compartmental model using an SEIR framework. We divided the population into states with respect to SARS-CoV-2 infection: susceptible (S); exposed but not yet infectious (E); infectious and pre-symptomatic ($I_P$), asymptomatic ($I_A$), mildly symptomatic ($I_M$), or severely symptomatic ($I_S$); hospitalized cases that will recover ($H_R$) or die ($H_D$); recovered and immune (R); and dead (D). We assumed an underlying, unobserved process model of SARS-CoV-2 transmission depicted in figure 2. We used a Euler approximation of the continuous time process with a time step of 4 h. We assumed that transitions between states were simulated as binomial or multinomial processes, which treat periods within each state as being geometrically distributed. Given that each period is geometrically distributed with a different rate, transition times through multiple states follow no named distribution but are unimodal (e.g. disease onset-to-death: electronic supplementary material, figure S1). It is

also possible to divide each state (e.g. infectious classes) into multiple sub-stages to produce Erlang-distributed periods within stages [45,46] (a change that we implemented in later iterations of our model [42]); however, here we relied on single compartments for each state for simplicity (electronic supplementary material, table S1). The equations (electronic supplementary material, Eq. S1–Eq. S8) describe in detail the stochastic transitions between states. We assumed that the observed deaths are a Poisson random variable with mean of total new deaths accumulated over the observation period (1 day for this analysis).

The transmission parameter, $\beta_t$, describes the average *per capita* contact rate, at time $t$, between susceptible and infectious people multiplied by the per-contact transmission probability. We defined $\beta_t$ as being directly proportional to the impact of social distancing at time $t$, which is given by $\sigma_t$, such that $\beta_t = \beta_0 \cdot \sigma_t$, where $\beta_0$ is the transmission rate prior to any social distancing (with $\sigma_t = 1$). The degree to which a social distancing intervention reduces the overall population contact rate is $1 - \sigma_t$. While any sequence of time-varying transmission rates can be implemented in this framework, given the limited data to inform estimates of different transmission rates, we model $\beta_t$ in three distinct segments of time that are characterized by different social contact structures: (1) baseline prior to any interventions ($\beta_0 = \beta_0 \cdot 1$), assumed to occur at least until 29 February (2) the San Francisco Bay Area 'work-from-home' initiative, which we model as beginning some time between 1 March and 9 March ($\beta_{WFH} = \beta_0 \cdot \sigma_{WFH}$); and (3) the San Francisco Bay Area shelter-in-place, which began on 17 March ($\beta_{SIP} = \beta_0 \cdot \sigma_{SIP}$). We included $\sigma_{WFH}$ and the work-from-home start date as two of the parameters we sampled across a plausible parameter range (see electronic supplementary material, table S4) but allowed $\sigma_{SIP}$ to be estimated by the model (see 'Fitting the Model' below). We did not fit $\sigma_{WFH}$ due to concerns about identifiability as $\sigma_{WFH}$ modifies the transmission rate for only a brief period of time prior to the first observed death in Santa Clara County (figure 1).

By including asymptomatic and pre-symptomatic individuals, we were able to track 'silent spreaders' of the disease, both of

which have been shown to contribute to COVID-19 transmission [47] (electronic supplementary material, table S1). Tracking deaths allowed us to compare our simulations to a data source that was likely more reliable than confirmed cases, particularly in the absence of widespread rapid testing and case detection. Mildly symptomatic cases were defined as those people that show symptoms but do not require hospitalization, while we assumed that all severely symptomatic cases would eventually require hospitalization (figure 2). We also assumed that no onward transmission occurred from hospitalized individuals. We further assumed that all individuals not exposed to the virus begin as susceptible to infection, and that all model compartments other than susceptible and exposed began with zero individuals.

## (b) Fitting the model

We fit $\beta_0$, which describes the initial transmission rate in the absence of any interventions; $\sigma_{SIP}$, which describes the proportional reduction in $\beta_0$ under shelter-in-place; and $E_0$, with which we drew the initial number of exposed individuals as Poisson($E_0$) + 1. Fitting more than these parameters with only a few weeks of daily deaths (our first model iteration was hosted online on 22 March 2020: figure 1), was unrealistic because parameters were not identifiable. To estimate these three parameters, we assumed point estimates for parameters for which there was at least some convergence in estimates in the literature (electronic supplementary material, table S3); most notably these parameters include the average time individuals spend in infectious states. We use the inverse of durations as average exit rates, but note the possibility that taking the inverse of durations from individual-based studies (e.g. incubation period [49] and time from symptom onset to death [59,62]) might not scale appropriately for use as rates in a population-level model. For the remaining parameters, we drew 200 Sobol sequences, a more efficient method than Latin hypercube for sampling input parameters [64], across a range of plausible values (electronic supplementary material, table S4) to form 200 plausible parameter sets. While sampling over all non-fitted parameters is possible, we decided against this strategy in an effort to focus computation time on the areas of greatest uncertainty.

We note that we use $\lambda$ to refer to the exponential rates at which individuals leave infectious classes and not force of infection as is common. The $\kappa$ parameters (electronic supplementary material, table S3) scale $\beta_t$ for individual infectious classes, where $\kappa = 1$ indicates no scaling. For simplicity and in the absence of better data, we assumed that only asymptomatic transmission had a scaling factor different from one (specifically, $\kappa_A < 1$; see electronic supplementary material, table S4). For the rates shown in electronic supplementary material, tables S3 and S4, the unimodal distribution for the time from first symptoms to death has a mean of approximately 23.5 days, a median of 20 days, a mode greater than zero and a moderate right-skew (electronic supplementary material, figure S1). This median is between the mean value of 17.8 found by Verity *et al.* [59,72] and the range of 35–44 days observed in Bi *et al.* [73].

Using the pomp (statistical inference for partially observed Markov processes) package [74] (function mif2) in the R programming language [75], we fit $\beta_0$, $\sigma_{SIP}$ and $E_0$ to daily deaths for each of the 200 parameter sets using six independent replicate particle filtering runs. For each independent replicate, we perturbed the starting values for fitted parameters among runs using random samples from a lognormal distribution for $\beta_0$ (meanlog = log(0.7), sdlog = 0.17) and $\sigma_{SIP}$ (meanlog = log(0.2), sdlog = 0.2) and a uniform distribution between 0 and 6 for $E_0$. Each individual mif2 replicate run used 300 iterations, 1000 particles, a cooling fraction of 0.50, and a random-walk perturbation for all parameters of 0.02 (using the function ivp for $E_0$ to designate it as an initial value parameter). The optimization was constrained to positive values for $E_0$ and $\beta_0$ and between zero and one for $\sigma_{SIP}$. After the

filtering steps were completed, for each mif2 replicate run, log-likelihoods were calculated using the function pfilter 10 times with 10 000 particles each to produce both mean and standard errors for log likelihoods for each parameter set.

We computed weekly fits from 1 April through 24 June by withholding data reported after the given fit date. We used COVID-19 death data from *The New York Times*, based on reports from state and local health agencies (available at https://github.com/nytimes/covid-19-data, figure 3b). Daily deaths were calculated from differences in cumulative death reports. Using these data, which are available for all counties in the USA, our model can be used to fit $\beta_0$, $\sigma_{SIP}$ and $E_0$ in any county.

We calculated $\mathcal{R}_0$ as estimated $\beta_0$ times the duration and infectiousness of an average infection (as defined by our model structure) for each of the 1200 parameter sets (using all six estimates from each of the mif2 replicate runs). For each of the 1200 parameter sets, we calculated the effective reproduction number $\mathcal{R}_E$ for each weekly fit by modifying the calculation for $\mathcal{R}_0$ to scale the estimated $\beta_0$ by both $\sigma_{SIP}$ and the estimated median proportion of the population remaining susceptible across 200 simulated epidemics for the given parameter set.

For each fitted model parameter ($\beta_0$, $\sigma_{SIP}$ and $E_0$), uncertainty comes from two sources: variation in fitted values among replicate mif2 iterations (e.g. uncertainty in the value of fitted parameter conditional on a given parameter set—a single given conceivable state of the world), and variation in the estimated parameter value across the 200 parameter sets (uncertainty in the value of the fitted parameter given uncertainty in the state of the world). We computed likelihood profiles for the three fit parameters over 30 uniformly spaced points (hereafter, fixed points): from 0.2 to 1.2 for $\beta_0$, 0.01 to 0.6 for $\sigma_{SIP}$ and 1 to 30 for $E_0$. For each parameter and each fixed point, we refit the model for each of 200 unique Sobol-sequenced parameter combinations with the same mif2 settings used in other model fitting steps (except with three mif2 replicates rather than six due to computational cost). We identified the maximum log-likelihood for each fixed point among all 600 fits (200 parameter sets, each fit three times with random starting values). We computed likelihood profiles for three fit dates only (1 April, 13 May and 24 June 2020) because of computational costs (18 000 model fits per profile).

## (c) Simulating epidemics

While a set of parameters can produce many simulations, only some of these trajectories are conceivable given the observed data. To simulate using only trajectories that are plausible conditional on the observed data, we drew trajectories from the smoothing distribution using the filter.traj and pf functions in pomp with 5000 particles for each particle filter. These filtering trajectories can be viewed as weighted samples from the distribution of unobserved state processes given the observed data, where the weights are determined by the likelihoods from the particle filtering [74]. All simulations were run forward in time from filtering trajectories, which constrains forecasts to continue from a present state matched to the observed epidemic dynamics in order to avoid overly large forecast uncertainty. Unless otherwise noted, all simulations used parameter sets within the top two log likelihood units for each fit date. For all simulated trajectories, we used 25 filtered trajectories and 25 forward simulations from each filtered trajectory for a total of 625 total epidemic forecasts for each of the parameter sets within the top two log-likelihood units for each fit date (the number of which varied by fit date). Unless noted otherwise, the uncertainty bands that we display for all simulations prior to the fit date contain the central 95% range of outcomes across parameter variation among the fits within the top two log likelihood units and variation among the 25 filtered trajectories. The uncertainty bands after the fit date (forecasts) contain the same parameter variation

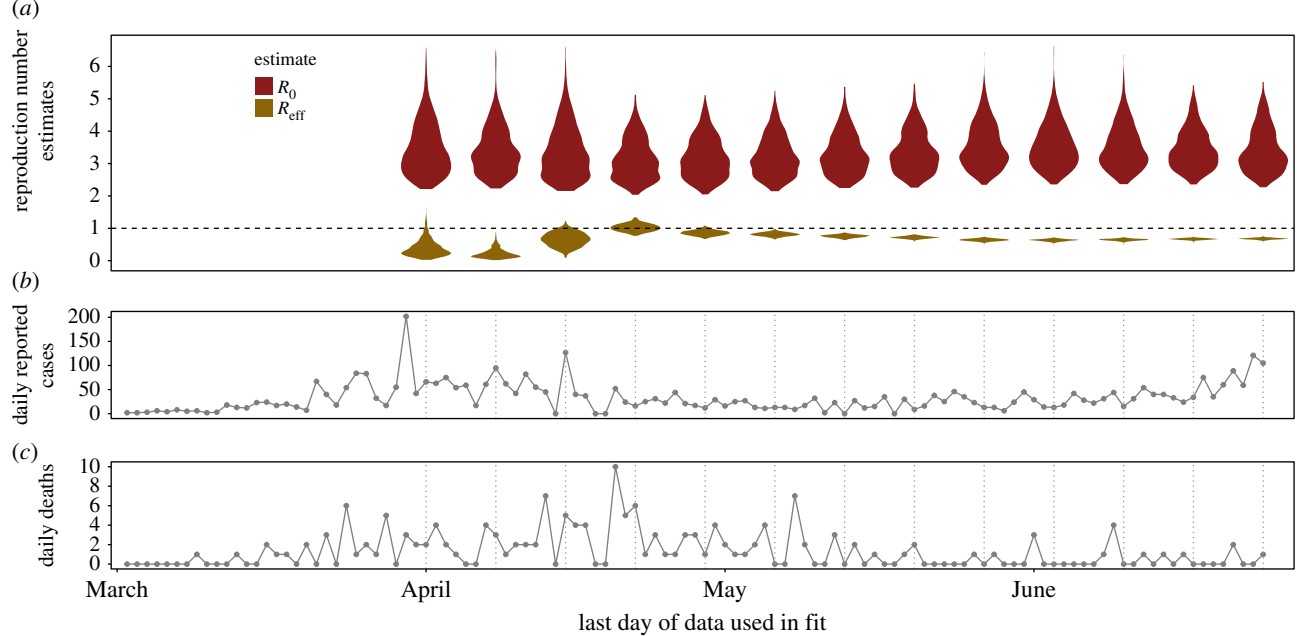

**Figure 3.** $\mathcal{R}_0$ (red, upper) and $\mathcal{R}_E$ (gold, lower) estimated using time series death data up to the date on the horizontal axis (violin density plots, $a$). Time series show reported cases ($b$; not used in model fitting) and reported deaths used for model fitting ($c$) in Santa Clara County, with vertical dotted lines marking last data point used in each corresponding model fit ($a$). (Online version in colour.)

and stochastic variation and include additional variation from 25 forward simulations for each of the 25 filtered trajectories. The uncertainty bands on future simulations of deaths also contain additional variation from the Poisson observation process. These are simultaneously wide because of large numbers of stochastic simulations, but narrow because we ignored uncertainty in the parameters listed in electronic supplementary material, table S3 as well as uncertainty in the estimated parameters for each fit, and thus should be interpreted with caution.

### (i) Model assessment

To assess model performance, we compared model-forecasted deaths—simulated for 14 days forward in time from filtering trajectories assuming that the existing levels of social distancing were maintained—to observed deaths. Specifically, we quantified model forecast performance for deaths using the 'quadratic score'. The quadratic score is a commonly used strictly proper scoring rule (a forecasting evaluation metric with a unique maximum that is reached by increasing both the accuracy and the concentration of the predictive distribution around the true value) for a predictive model with a discrete (e.g. Poisson) error distribution [76–79]. We calculated the quadratic score for each simulation over the 14 days following the fit date as

$$\frac{1}{N}\sum_{i=1}^{N}[-2(Pr(y_i|\hat{\mu}_i)) + \sum_{k=0}^{\infty}(Pr(k|\hat{\mu}_i))^2],$$

where $i$ indexes the days since the fit date (from 1 to $N = 14$), $y_i$ is the observed new daily reported deaths, $\hat{\mu}_i$ is the daily prediction from the simulation, and $Pr(y_i|\hat{\mu}_i)$ is the probability mass on $y_i$ given the prediction $\hat{\mu}_i$. The sum $\sum_{k=0}^{\infty}(Pr(k|\hat{\mu}_i))^2$ runs over all positive integers ($k$) to measure the dispersion of probability mass given the prediction $\hat{\mu}_i$; for a Poisson distribution, we calculate this sum analytically based on the following closed form expression for any $x \geq 0$: $\sum_{k=0}^{\infty}(Pr(k|x)^2) = e^{-2x}I_0(x)$, where $I_0$ is the Bessel function. For the model predictions $\hat{\mu}_i$, we used the underlying new daily deaths from the simulation's trajectory ($D_{new}$), rather than the deaths arising from the Poisson observation process, given that the model-predicted distribution of deaths for a given simulation is best characterized by Poisson($D_{new}$). For each fit date, we show the distribution of

quadratic scores across simulations from fitted parameter sets within the top two log-likelihood units (one score per simulation and 625 simulations—25 forecasts for each of 25 filtered trajectories—per fitted parameter set).

We refrained from calculating a quadratic score comparing the model's predicted cases and the reported cases given that we did not model incomplete case detection. Instead, we simply relied on a visual comparison between the trajectory (curvature) of the predicted cases and reported cases to qualitatively assess the model's predictive accuracy. Specifically, we compared predicted new symptomatic infections to the cases reported one week later in order to account for a week of reporting lag.

### (ii) Future interventions

Our modelling framework allows for different types, intensities and durations of interventions, and thereby illustrates how these interventions impact dynamics and the resulting number of COVID-19 cases and fatalities through time. Here, we use fits generated from deaths reported prior to 22 April to consider three possible interventions that can be implemented at different times during the simulation:

1. *Social distancing for a set duration* applied as a scaling of the transmission rate for all individuals;
2. *Isolation of symptomatic individuals* applied as a scaling of the transmission rate for only symptomatic individuals $I_S$ and $I_M$; we assume isolation paired with partially relaxed social distancing;
3. *Adaptive triggering* applied as a tightening or relaxing of social distancing, triggered by hospitalizations crossing a defined threshold.

Other scenarios that can be modelled as a time-varying reduction in $\beta_0$ (such as contact tracing and quarantine, which we do not include here) can be explored using the open-source code (available at https://github.com/marissachilds/COVID19_early_model) and saved model fit data files (available on Dryad: https://doi.org/10.5061/dryad.cvdncjt4t).

To visualize the dynamics of a single intervention scenario, we simulated 25 epidemics from each of 25 filtered trajectories

(for a total of 625 total simulations) from the parameter set with the best likelihood using data up to 22 April. To quantify the effectiveness of each intervention scenario, we estimated cumulative deaths and the number of new symptomatic infections over time from the simulated epidemics for parameters sets within 2 log-likelihood units of the best for each intervention (e.g. the effectiveness of infected isolation).

### (iii) Counterfactuals

In addition to making forward projections under different intervention scenarios, models can help compare past actions taken (e.g. public health interventions) to alternative hypothetical scenarios (e.g. alternative types and timings of non-pharmaceutical interventions). Such comparisons can help to highlight which actions were the most helpful and which could have been improved. Early in an epidemic, a counterfactual analysis is particularly useful to assist local policymakers and the public contextualize the impact of early decisions relative to other possible decisions that could have been made. To assess the impact that existing county orders and resources had on the epidemic trajectory, we limited filtering trajectories to the date in which the counterfactual scenarios diverged (i.e. 17 March when the county shelter-in-place order went into effect), then simulated forward assuming: (1) shelter-in-place orders went into effect on 17 March; (2) shelter-in-place orders went into effect one week later on 24 March and (3) testing and isolation of infected individuals began in addition to the shelter-in-place orders on 17 March (in reality, testing remained limited in Santa Clara County and throughout the USA through the end of April [80]). In particular, we assumed that testing and isolation of symptomatic individuals further reduced their infectious contacts by 80% for severely symptomatic individuals and 70% for mildly symptomatic individuals.

## 3. Results

### (a) Model estimates and performance over time

We iteratively fit our model each week from 1 April through 24 June 2020 using data on daily reported COVID-19 deaths up to that date. For each fitted model, we first estimated $\mathcal{R}_0$ and $\mathcal{R}_E$ to investigate how our understanding of epidemic dynamics changed with increasing data availability, then compared the fit of simulations to out-of-sample data to evaluate how model performance changed over time.

The model consistently estimated that $\mathcal{R}_0$ was between 3 and 4 (with a median among all fits of 3.76), though $\mathcal{R}_E$ varied considerably, especially among the fits throughout April (figure 3). For example, $\mathcal{R}_E$ jumped from a confident estimate below one on the 15 April fit to an uncertain estimate spanning one on the 22 April fit, after a week of higher deaths was included. After the April volatile period, $\mathcal{R}_E$ estimates stabilized near 0.69 by mid-late May with very narrow confidence intervals (e.g. on 27 May the model estimated a median $\mathcal{R}_E$ of 0.642 with a 95% CI of 0.571–0.708 among fits within two units of the top log likelihood) and continued to vary little throughout June. Even as cases increased in June (figure 3), deaths remained low, leading to little change in $\mathcal{R}_E$ estimates. The estimated impact of social distancing ($\sigma_{SIP}$) and transmission rate in the absence of non-pharmaceutical interventions ($\beta_0$) followed a similar pattern of increased confidence over time, while the initially exposed class ($E_0$) was estimated with large uncertainty in all fits (electronic supplementary material, figures S2 top panel and S3). Profiles over the fitted parameters reflect these patterns: the $\beta_0$ and $\sigma_{SIP}$ profiles showed clear peaks, the $E_0$ profile was flat across

the range of values examined, and the $\sigma_{SIP}$ profile began showing jaggedness for the last fit date (24 June 2020) when the model was no longer suited for the changing epidemiological situation (electronic supplementary material, figure S2 bottom panel). The difficulty in identifying the initial value parameter $E_0$ is not surprising given that only early time points are expected to inform the initial state in a stochastic process [81]; in the case of this dataset, the time between the initial conditions (the start of the epidemic) and the first observation reached up to 68 days for some parameter sets. We ran additional diagnostics to understand the effect of the model's difficulty in identifying $E_0$ (electronic supplementary material, figure S3) on the other focal parameters and quantities of interest. We found that altered mif2 settings (an expanded range of starting values and random walk standard deviations; see electronic supplementary material, figure S3): (1) permitted larger $E_0$ values which allowed for overly late, biologically implausible, start dates (electronic supplementary material, figure S3A); (2) did not lead to much better convergence (electronic supplementary material, figure S3B) and (3) despite allowing for much larger estimates of $E_0$, did not meaningfully impact the values of other fitted parameters (electronic supplementary material, figure S3C).

Despite the uncertainty in initial conditions, strong convergence of the 10 replicate log-likelihood estimates (as measured by small standard errors among them, electronic supplementary material, figure S4), the six replicate mif2 runs in terms of log-likelihood (electronic supplementary material, figure S5), and good convergence in estimates for $\sigma_{SIP}$ and $\beta_0$ (electronic supplementary material, figure S6) indicates that much of this uncertainty is due to the inability of our model to differentiate among alternative parameter sets (parameter identifiability issues) (electronic supplementary material, figures S7–S10) and not a misuse or failure of the fitting algorithm or log-likelihood calculations.

Near-term forecasts and model performance also varied substantially over time. Simulations based on the model fit to deaths through 1 April show that while the bulk of simulations predicted declining deaths (figure 4, corresponding to the majority of the $\mathcal{R}_E$ density being below 1, see figure 3), some simulations show rapidly increasing daily deaths over time (corresponding to an $\mathcal{R}_E > 1$). With the 22 April model fit, uncertainty in whether $\mathcal{R}_E$ was above or below 1 (figure 3) resulted in very large uncertainty in epidemic trajectories (figure 4). Simulations from later model fits (e.g. those from late May through early June) projected a decline in deaths through the end of June; model fits in June consistently suggested epidemic fade-out by the end of the month, assuming that the existing non-pharmaceutical intervention regime had remained in place. Weekly model fits (electronic supplementary material, figure S11) show that the model tended to under-predict deaths early in the study period (early to mid April) when a period of low daily deaths led to low estimates of $\mathcal{R}_E$ (figure 3, gold shaded violin plots). The model then equally under- and over-predicted deaths in the middle of the study period (late April to mid-May), when sufficient data had likely allowed for more accurate estimates of the shelter-in-place effectiveness. Finally, the model under-predicted deaths again at the end of the study period (mid-May to late June) after shelter-in-place orders were relaxed on 4 May (figure 1) and the single estimated value for the effectiveness of social distancing was no longer realistic.

The model qualitatively matched the curvature in the cases reported within the time window used to fit the model for fits through 13 May (figure 5, blue points), although, as expected, the model predicted far more new symptomatic infections than reported cases (figure 5, difference in left and right vertical axes). Estimates of daily cases through the date of fitting produced realistic estimates of the proportion of the population remaining susceptible (electronic supplementary material, figure S12). However, all future projections failed to capture the trajectory of future, out-of-sample, reported cases; in most instances the model predicted declining cases despite the increase in cases observed starting in late May. By 3 June, the model began to fail to capture the increasing cases even within the observed time period for which death data were used to fit the model (figure 5, blue points). This inaccuracy in predictions was likely due to both the changing epidemiological environment (figure 1) that made model assumptions unrealistic (i.e. a constant effectiveness of social distancing, $\sigma_{SIP}$, and constant mortality rate) and the fact that changes in epidemic dynamics will be apparent in cases prior to deaths.

## (b) Scenario analysis

We originally designed and fitted our model (and accompanying interactive website) in part to communicate to local policymakers and the public the impact of the early social distancing interventions in Santa Clara County and the importance of continuing strong non-pharmaceutical interventions for saving lives and preventing an epidemic resurgence. To achieve these goals, we used counterfactual analyses to compare what transpired to alternative unrealized scenarios and to forecast the epidemic under alternative future scenarios with different non-pharmaceutical interventions. We revisit these analysis here, in brief, to illustrate this use of our model.

We estimated that a second peak would have been inevitable in the absence of any non-pharmaceutical interventions even if shelter-in-place had been maintained until 1 June 2020, as illustrated here for the single best-fitting parameter set (figure 6, red lines). Across all parameter sets within 2 log-likelihood units of the MLE and stochastic epidemic simulations, we estimated that in this scenario Santa Clara County would have had a median of 6140 deaths (95% CI: 546–19 494) and a peak number of daily new symptomatic infections of 33 193 (95% CI: 12 536–58 259) occurring on 5 July 2020 (95% CI: 26 June 2020–18 July 2020).

Maintaining shelter-in-place until 1 June, followed by less stringent social distancing (50% of baseline contacts), combined with strong symptomatic case isolation (removing an additional 80% and 70% of contacts from severe and mild symptomatic infections, respectively), would have allowed for higher background contact rates (e.g. more businesses reopening) and yet fewer deaths, as predicted by our single best model fit (figure 6). For a range of possible combinations of symptomatic case isolation efficiencies and background social distancing (electronic supplementary material, figure S13), we found an overlap in confidence intervals for deaths, but higher median estimated deaths at the weakest levels of social distancing in the general population. For reference, the median number of estimated deaths under maintained shelter-in-place is shown by the horizontal black line, with 80% and 95% CI in dashed and dotted lines, respectively (electronic supplementary material, figure S13). These confidence intervals span a wide range because our estimated $\mathcal{R}_E$ values

as of 22 April ranged from 0.76–1.34, which led to some epidemics growing and some declining through time.

We proposed that, without widespread testing availability before the end of shelter-in-place, a hypothetical alternative strategy would have been adaptive triggering, in which social distancing orders are intensified and relaxed as hospitalizations exceed and fall below critical thresholds, respectively. However, because the estimated $\mathcal{R}_E$ for Santa Clara County was approximately one (and the confidence interval included one on 22 April), a strategy that periodically reduces the strength of social distancing may have led to an overall increase in cases that would not be reversed when the shelter-in-place was reinstated. We found that an adaptive triggering strategy that alternates between social distancing that reduces transmission to 20% and 80% of baseline could be effective in keeping cases and deaths low (electronic supplementary material, figure S14). This method would have kept the epidemic within the capacity of the healthcare system, but resulted in prolonged cycles of epidemic resurgence and control, continuing until herd immunity was reached through recovery of infected individuals or vaccination.

In simulations of counterfactual scenarios, we found that an additional 57 (95% CI: 10–143) lives would have been lost if shelter-in-place orders had been delayed even a week, and 26 (95% CI: 3–51) deaths could have been averted if testing and isolation of symptomatic individuals was available from the time of the shelter-in-in place (electronic supplementary material, figure S15).

## 4. Discussion

During an unfolding pandemic, modelling is an essential tool for tactical decision-making, strategic planning and communicating qualitative scenarios [17]. Many early COVID-19 models played a critical role in highlighting the importance of social distancing to governments and to the public (e.g. [2–5]). Models like our own helped communicate to the public that 'flattening the curve' and slowing transmission was not a short-term endeavour, but also that early and sustained interventions had major benefits for local public health. Despite their importance, it is often unclear how quickly and for what reasons early models become obsolete given that retrospective analyses are not usually conducted on the first iterations of models. Here, we presented an example retrospective analysis on our early COVID-19 model to ask the following questions: (1) what did the model suggest about epidemic metrics, dynamics, and the impact of non-pharmaceutical interventions, and how did these estimates change over time? (2) How accurately did the model predict epidemic dynamics going forward? (3) For how long was the model accurate enough to be useful, and what limited its longer-term accuracy?

## (a) Epidemic dynamics over time (Q1)

Our model stably estimated $\mathcal{R}_0$ between April and June with an overall median of 3.76 (figure 3), comparable to values estimated elsewhere (e.g. [82]), and identified that $\mathcal{R}_E$ declined substantially after the shelter-in-place order was enacted, to near or below one. However, predicted future epidemic dynamics were highly uncertain given that many model parameters had large uncertainty (especially for the 1 April model fit, see electronic supplementary material, figures S2 and S7), estimated credible intervals on $\mathcal{R}_E$ spanned one in

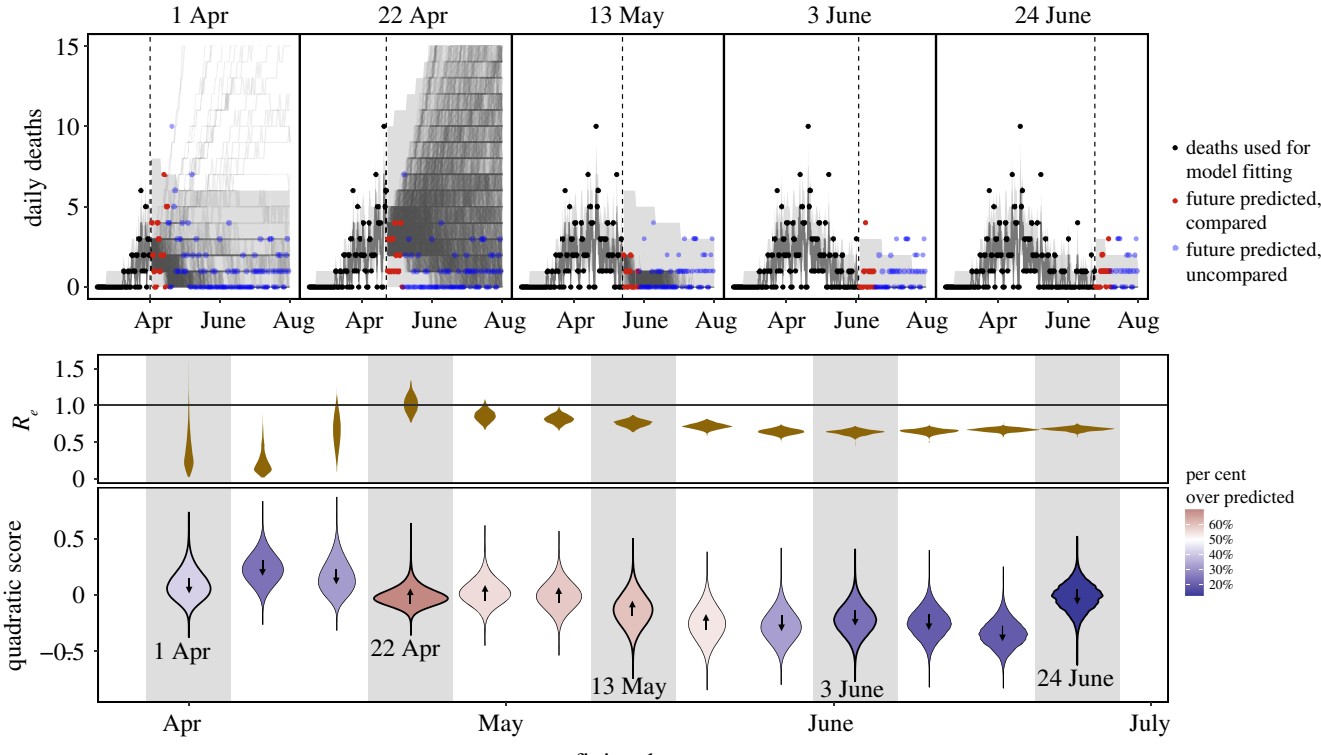

**Figure 4.** Model accuracy at predicting deaths over time peaked in mid-May to early June, as $\mathcal{R}_E$ stabilized near 0.69. Top panels are simulations of the model fitted to reported deaths (black points) up to dates spanning from 1 April (left) to 24 June (right) and compared to future reported deaths (red points for deaths within two weeks, which were used for quadratic scoring, blue points for future deaths beyond two weeks). For future simulations (right of dashed line), medians of simulated observed deaths for each parameter set are shown as lines and a 95% confidence interval over all simulations from all parameter sets are shown in grey ribbons. For past filtered trajectories (left of dashed line), grey lines show medians of new deaths based on the underlying state process filtering trajectories (i.e. without a Poisson observation process) for each parameter set and grey ribbons show 95% central intervals among all trajectories for all parameter sets. Gold violin plots (middle) show the distribution of $\mathcal{R}_E$ values on each fit date; violins on a grey background correspond to the trajectories in the top panels (matched by date). Blue-to-red coloured violin plots (bottom) illustrate the distribution of the quadratic score across simulations for the two weeks following each model fit. A lower quadratic score reflects better model predictions. Red shading and an up arrow indicate a higher percentage of overestimates (above 50%) and blue shading and a down arrow indicate a higher percentage of underestimates. All parameter sets within two log-likelihood units are included for each each fitting date, with 25 filtered trajectories for each parameter set and 25 forward simulations for each filtering trajectory resulting in 625 simulations for each parameter set. (Online version in colour.)

model fits until early-mid-May (figure 3), and $\mathcal{R}_E$ estimates were variable from week to week when data were sparse (figure 3). For example, the inclusion of a week with higher deaths (most of which occurred in long-term care facilities [83], a distinction that is not captured in our model that assumes a homogeneous population), led estimated $\mathcal{R}_E$ to jump to span one on 22 April (figure 3). The volatility and uncertainty in $\mathcal{R}_E$ estimates highlight the difficulty in inferring epidemic metrics from deaths alone, which are a noisy and lagged indicator of the underlying epidemic dynamics. Despite these limitations, early models like ours play a critical role in estimating coarse epidemiological metrics and dynamics (e.g. $\mathcal{R}_0$ and $\mathcal{R}_E$), which are fundamental for quantifying and comparing the efficacy of various intervention scenarios and predicting the future course of the epidemic.

### (b) Forecasting accuracy (Q2)

Our model initially gained accuracy in predicting epidemic dynamics as additional data increased parameter identifiability (April to mid-May 2020, figure 4, bottom panel) but then began to decline in performance as the model assumptions became too simplistic to capture the changing epidemiological context (late May to June 2020). Out-of-sample predictions of deaths suffered from high uncertainty during the early period (e.g. on 22 April)

while predictions during the end of the study window (e.g. on 3 June) became overly confident that the epidemic would die out. Because of limited and variable testing capacity, we relied on a visual comparison of the curvature in model predictions of new symptomatic infections to observed daily cases (lagged one week for a plausible reporting lag). Model fits through 13 May qualitatively match the curvature in the reported cases within the time window used to fit the model, but by 3 June the model failed to capture even the increasing cases within the observed time period (figure 5, blue points). Thus, our model illustrates that trajectories of cases can be captured in relatively simple mechanistic models based on only reported deaths, but only for a limited period of time until epidemiological conditions change.

### (c) Limitations to long-term accuracy (Q3)

Predictions deteriorated as the epidemiological environment (figure 1) began to deviate further from our model assumptions. We fit a simple step function for the impact of non-pharmaceutical interventions with the aim of balancing identifiability in the face of limited data and accuracy of early predictions. However, this assumption restricted the utility of our model once Santa Clara County began to relax social distancing orders (figure 1). Additionally, the

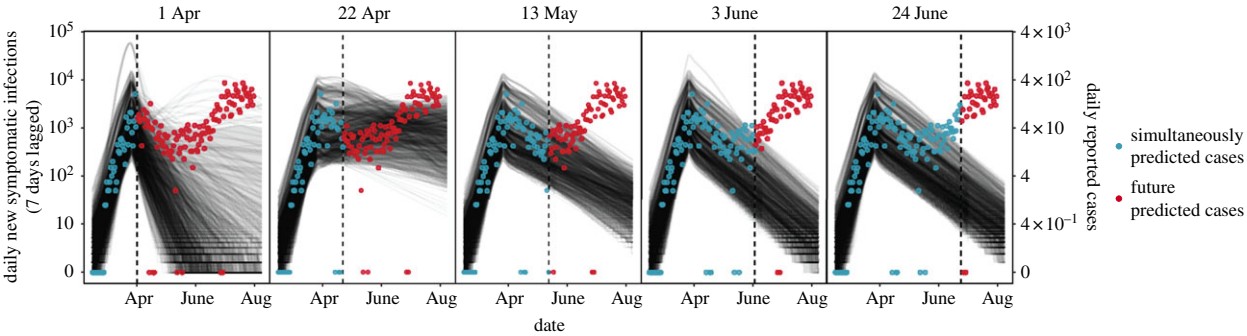

**Figure 5.** The model accurately predicted cases until early June, when social distancing orders were relaxed. Panels show simulations of the model (which is fitted to reported death data; not shown) from 1 April (left) to 24 June (right), on a log scale. Model simulations of daily new symptomatic infections (with a one-week lag to approximate a plausible time from symptom onset to case reporting) are shown with lines and correspond to the left *y*-axis. Note that reported cases and predicted symptomatic infections differ by approximately 2 orders of magnitude, as expected due to under-reporting. As in figure 4, each black line represents the median among simulations for a single parameter set and the grey shaded region represents the 95% central interval among all simulations. Reported cases are shown as points and correspond to the right *y*-axis: blue points are cases reported during the time period used in model fits, while red points (those to the right of the vertical dotted line) are cases reported after the time period associated with model fitting. The vertical dashed line indicates the date of model fitting. (Online version in colour.)

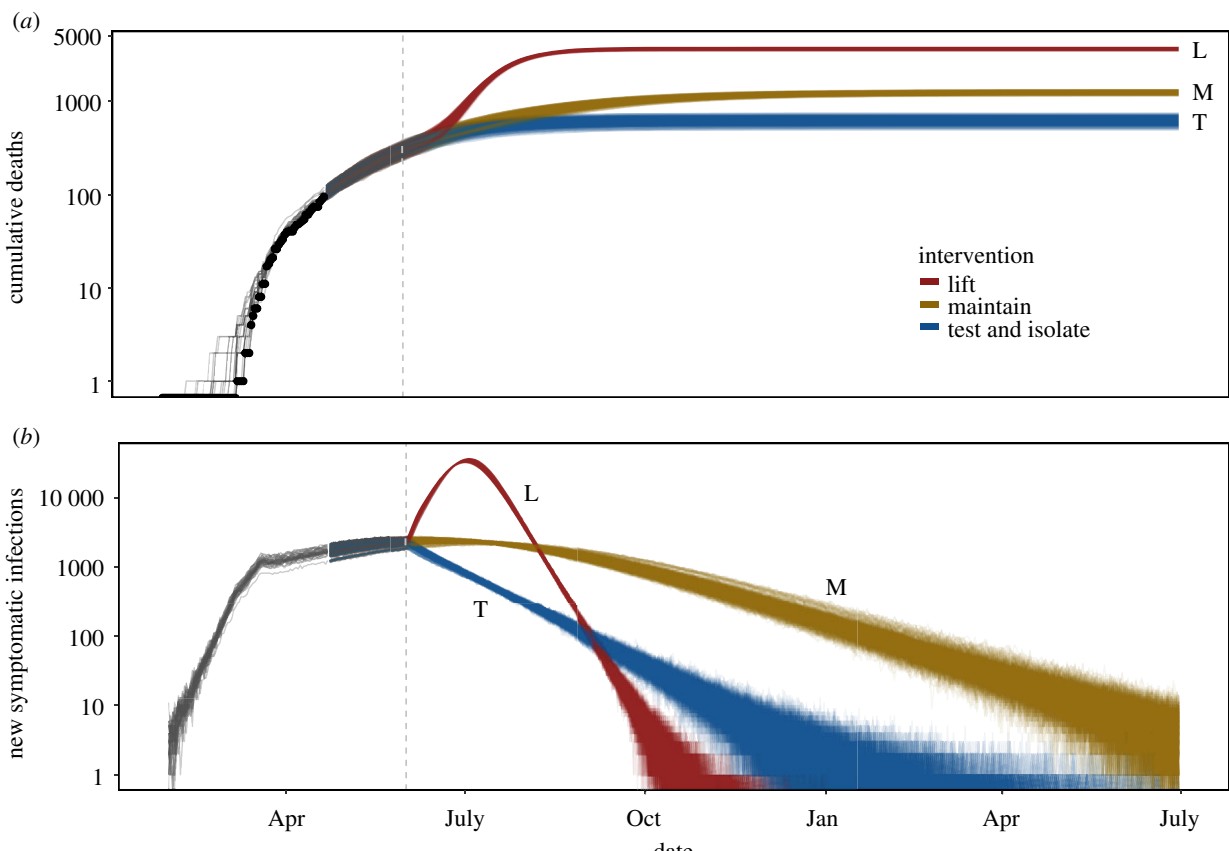

**Figure 6.** Maintaining non-pharmaceutical interventions after 1 June, when early shelter-in-place restrictions were relaxed, is critical for preventing a devastating resurgence. Maintaining shelter-in-place at the $\sigma_{SIP}$ value predicted by the model (here $\sigma_{SIP} = 0.40$) (gold) or test-and-isolate ($\sigma_{SIP} = 0.50$ plus an additional 80% effectiveness of test and isolate for severe infections and 70% for mild infections) (blue) strategies over long periods are necessary to prevent a major epidemic resurgence (red) following the end of the initial shelter-in-place order on 1 June (dashed vertical line) in Santa Clara County. Grey lines prior to 22 April show 25 filtered trajectories; coloured lines after 22 April show 25 simulated trajectories from each of the 25 filtered trajectories for each of three intervention scenarios. Black points in *a* show observed data. Cumulative deaths are plotted for visualization only; model is fit using raw death counts. For ease of visibility, only simulations from the parameter set with the maximum likelihood from the 22 April fits are shown here. (Online version in colour.)

relationship between cases and deaths fundamentally changed between the first and second waves. COVID-19 deaths remained relatively flat through July and August even while reported cases surged above 300 per day by the third week in July [83]. This pattern occurred in many places in

the US during the summer resurgence [84] and may reflect some combination of differences in personal protective behaviours and social distancing adherence across disease severity risk groups, resulting in a larger share of cases occurring in people less vulnerable to severe disease and death. Improved

standards of care, safety protocols and the availability of personal protective equipment in long-term care facilities, as well as increased testing to improve the detection of asymptomatic or mildly symptomatic cases all likely played a role as well.

## (d) Early modelling and lessons for future epidemic control

Generally, real-time models face opposing forces: additional time means additional data to aid in parameter estimation; however, as more time passes the epidemiological environment deviates further from that which the early model was built to address, eventually reducing model accuracy [17,19]. Our case study and others (e.g. [85]) clearly demonstrate these opposing forces. Rapidly changing US policy forced many COVID-19 models to consider continuous-time estimates of movement (e.g. [42,86–89]) instead of constant intervention strengths. Yet, even these more detailed models faced a changing epidemiological context, which included the implementation of mask mandates and other behavioural changes (e.g. [90,91]), seasonality modifying viral kinetics and behavioural contact patterns, holidays altering mixing and travel patterns, and emerging virus variants with new epidemiological properties. In general, changes such as these cause unexpected variation in disease dynamics across time and space, limiting the accuracy of long-term forecasts [13,17,19,20] and potentially reducing the accuracy of our model, which relied partially on fixed parameters derived from early outbreaks in other locations (e.g. China [51,57,61,66], Italy [44] and Singapore [50]).

While simplifying assumptions are necessary in early and real-time models, those assumptions must be frequently re-evaluated to ensure continuing accuracy [11,19,22,24,27,32,33]. We made a series of simplifying assumptions for our early model in an attempt to overcome a lack of data (electronic supplementary material, table S1). While we relaxed some of these in a follow-up model [42], several alternative technical decisions could have improved our model from the beginning. For example, we used single compartments for each state rather than subdividing the infectious states into multiple compartments, resulting in geometrically distributed transition periods instead of more realistic Erlang-distributed periods [45,46]. Though it saved us only one parameter, we also assumed a Poisson observation process rather than the more flexible negative binomial observation process which is often estimable without too much difficulty (e.g. [25,72]). Finally, we did not consider uncertainty in the estimated parameter values, for example by using importance sampling [92].

Our case study highlights key lessons for the practice of early, real-time modelling of emerging epidemics. In our analysis, we focused on quantifying near-term forecast accuracy, a common strategy for evaluating model performance [85,93], which could be used more frequently from the outset to monitor changes in model performance over time. We also advocate tracking variation in parameters and signs of declining parameter identifiability. Together, systematically applying these approaches will provide warning signs of model inaccuracies sooner and help to ensure that hidden mis-specifications are identified and corrected promptly. Early models will inevitably be imperfect reflections of reality constrained by limited data; using them responsibly requires considering and communicating uncertainty, a benefit of stochastic frameworks like that employed here [25]. Our model struggled to fit a relatively complex structure with a short time series of available data, particularly early in the epidemic, which resulted in some implausible epidemic dynamics (for example, unrealistically rapid depletion of susceptible individuals for some parameter sets from 1 April fits, electronic supplementary material, figure S12). This further highlights the difficulty of fitting complex models early in an epidemic and reinforces the importance of thorough evaluation of early predictions in order to avoid making biologically implausible claims. Our model was also particularly limited in its ability to estimate the initial number of infected individuals (electronic supplementary material, figures S2 and S3). In light of our and others' struggles with estimating $E_0$ for COVID-19 and other emerging infectious diseases [24,29,65], surveillance programmes (including genomic surveillance [94]) may assist with both disease control and modelling in early stages of an epidemic.

Regularly updated, centralized databases for parameter estimates (e.g. [95]) and relevant time series data on epidemiology and mobility (e.g. [41,96,97]) are major assets for modelling in emerging epidemics. These tools facilitate more rapid development of early models and streamline comparisons among models [16]. Changes to any model, especially those addressing flaws in previous versions, should be clearly communicated and shared with adequate documentation to ensure that outdated versions of the model are not used and do not guide others' model development [17]. However, especially in early stages of a pandemic, a relatively common and simple mechanistic modelling framework with a long and robust history, as outlined here, may be able to provide quicker and more reliable insight into disease dynamics than developing new model structures from scratch. For example, SIR-type models can do surprisingly well predicting epidemic metrics even with limited data [98–100]. Furthermore, simple transmission functions can be an effective alternative to more complicated functional forms, as illustrated here for the period of strong social distancing between approximately mid-March and mid-May 2020 (figure 1). Simple mechanistic models, unlike phenomenological models (e.g. statistical curve fits), also allow for scenario analysis through alterations in inputs and parameters, which allows for longer-term forecasts comparing alternative interventions.

Given the similarity between our model and others developed concurrently (e.g. [91]), we recommend developing infrastructure to facilitate collaboration, rapid communication, and workflows to minimize duplication of effort, facilitate troubleshooting, and aggregate and analyse projections across sets of models [101]. Further, we identified human behavioural changes as a key source of inaccuracy in our model predictions, suggesting the importance of collaboration between disease modellers and behavioural scientists, as well as guidelines for proper incorporation of mobility data [88]. Greater engagement between policymakers and scientists, particularly to clarify types and timings of interventions being considered, the importance of key modelling decisions, and the differences between early models considered in policymaking (e.g. the strengths and weaknesses of both phenomenological and mechanistic models) would ensure that a model is appropriately designed (e.g. distinguishing between symptomatic and asymptomatic infections to capture the dynamical implications of testing and isolating only symptomatic infections) and applied to relevant scenarios.

## (e) Conclusion

Despite all that has been learned about the impact of non-pharmaceutical interventions such as social distancing and mask wearing, and the approval of effective clinical therapies and vaccines, the US experienced two additional major epidemic waves within a year that each dwarfed the one in the early epidemic and control period we studied here. Given the order-of-magnitude difference in deaths and over three orders-of-magnitude difference in cases observed between the spring 2020 and winter 2021 periods [83], it may be tempting to conclude that non-pharmaceutical interventions and public health orders did not work, or were too economically and socially costly to justify their use. However, this is a dangerous conclusion. Mechanistic models like those we present here make it clear that, however imperfect, these interventions saved large numbers of lives: as of 1 July 2021, Santa Clara County has seen 2201 total deaths [83], a terrible toll, but one that is only 30% of our median prediction occurring from an unmitigated epidemic. As of July 2021, vaccination coverage in adults in Santa Clara County has reached approximately 75%, which, combined with the anticipated eligibility for younger children [102], portends an end to the epidemic locally in the coming months. Even during the peak of the winter surge, the county saw just over 700 concurrent hospitalizations, far shy of our median estimated value of 12 975 (95% CI: 760–28 927) that could have occurred without control measures in place. Although the US COVID-19 response clearly could have been better at controlling transmission, illness, and death, mechanistic models make it clear that the situation also could have been much worse without the control measures that remained in place, which were at least in part motivated by early models. Moving forward with COVID-19 and in future epidemics, models that incorporate changes in contact behaviour, population immunity derived from natural infection and vaccination, population heterogeneity in behaviour and immunity, and changes in immunity over time due to natural waning and emerging immune-evading variants will be critical for determining how to safely transition between initial and long-term interventions.

Data accessibility. Data used in this study are available at: https://github.com/nytimes/covid-19-data. Code used to produce the results in this study are available at: https://github.com/marissachilds/COVID19_early_model. Output from model fits and simulations are available from the Dryad Digital Repository: https://doi.org/10.5061/dryad.cvdncjt4t [103].

Authors' contributions. M.C.: conceptualization, data curation, formal analysis, investigation, methodology, software, visualization, writing-original draft, writing-review and editing; M.K.: conceptualization, data curation, formal analysis, investigation, methodology, software, visualization, writing-original draft, writing-review and editing; M.J.H.: conceptualization, data curation, formal analysis, investigation, methodology, software, visualization, writing-original draft, writing-review and editing; D.K.: conceptualization, data curation, writing-original draft; L.I.C.: conceptualization, data curation, writing-original draft; N.N.: conceptualization, data curation, writing-original draft; I.D.: conceptualization, data curation, writing-original draft; J.R.: conceptualization, software, visualization; A.D.B.: investigation, methodology, software, visualization, writing-review and editing; E.M.: conceptualization, funding acquisition, methodology, project administration, supervision, writing-original draft, writing-review and editing.

Competing interests. We declare we no competing interests.

Funding. Funding provided by: the National Science Foundation (DEB-1518681 and DEB-2011147, along with the Fogarty International Center); the National Institute of General Medical Sciences (R35GM133439); the Natural Capital Project; the Helman Scholarship; the Terman Award; and seed grants from the Stanford King Center for Global Development, Center for Innovation in Global Health, and Woods Institute for the Environment. M.L.C. was supported by the Illich-Sadowsky Fellowship through the Stanford Interdisciplinary Graduate Fellowship programme at Stanford University. M.P.K. was supported by the Natural Capital Project. N.N. was supported by the Stanford Data Science Scholars programme. J.R. was supported by the Terry Winograd Fellowship. M.J.H. was supported by the Knight-Hennessy Scholars Program.

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
