## [Peer Review File · Proceedings of the Royal Society B: Biological Sciences]

Review History

RSPB-2020-2281.R0 (Original submission)

Review form: Reviewer 1

Recommendation

Reject – article is not of sufficient interest (we will consider a transfer to another journal)

Scientific importance: Is the manuscript an original and important contribution to its field?

Acceptable

General interest: Is the paper of sufficient general interest?

Good

Quality of the paper: Is the overall quality of the paper suitable?

Marginal

Is the length of the paper justified?

Yes

Should the paper be seen by a specialist statistical reviewer?

Yes

Do you have any concerns about statistical analyses in this paper? If so, please specify them explicitly in your report.

Yes

It is a condition of publication that authors make their supporting data, code and materials available - either as supplementary material or hosted in an external repository. Please rate, if applicable, the supporting data on the following criteria.

Is it accessible?

Yes

Is it clear?

Yes

Is it adequate?

Yes

Do you have any ethical concerns with this paper?

No

Comments to the Author

The authors present a mathematical model of COVID-19 that accounts for non-pharmaceutical interventions. They used iterated filtering in the pomp R package to fit a stochastic discrete time version of their model. They evaluated the accuracy of the predictions of the model over time "to understand the value and limitations of models during unfolding epidemics."

The paper is well-written and interesting. The goal of the paper is very good. However, I have some concerns about the convergence of the parameter fitting using mif2. This convergence issue is always an important one when using stochastic fitting algorithms and it was not clearly shown by the authors that their parameters had converged. Here are some more specific comments:

Comments for Major Revision

1. Equations (1)-(10) are written in differential equation form where the transition terms $d_{X,Y}$ are transition rates. But the terms $d_{X,Y}$ as defined in equations (11)-(18) are defined for discrete time models. I understand that the continuous time stochastic process was simulated using Euler approximation. However the equations are inconsistent. There are not equations for the continuous time stochastic process. Equations (1)-(10) are written as if they are deterministic differential equations and Equations (11)-(18) are equations for discrete-time stochastic process. The authors need to choose a more consistent way to present their model.
2. Given the inconsistencies of Equation (1)-(10) and (11)-(18), I think it would be better to just put the diagram in the supplementary file in the main text and remove the differential equation system in (1)-(10) which the authors say they did not use.
3. The authors explain that they will use a transmission "parameter" β that will change over time. However this β does not appear in equations (1)-(10) or (11)-(18). I assume it is supposed to be part of the expression in equation (11) and was just accidentally dropped from there? This would be the normal location for it and what their online code suggests.
4. Regarding the parameter values chosen in Table 1. It is important to note that parameters estimated from other sources of data do not often translate to the same way in a population-level model. While I am not suggesting that the authors need to change their parameters, they should note that this can be an important issue. For example when using $\lambda_A=1/5$ days that

the authors chose for the asymptomatic rate (I should note that the authors actually wrote in Table 1 that $\lambda_A=5$ days which is inconsistent with equation (13) which clearly shows that λ_A should be a rate and not a period). Since the authors say in lines 74-75 that "We use an Euler approximation of the continuous time process," the binomial draws involving λ_A in equation (13) is supposed to approximate a continuous time process which means λ_A is supposed to be an exponential rate. However, I expect that most observations of the asymptomatic period does not have exponential distribution, but a more "central" distribution, such as an Erlang distribution.

5. In Tables 1-2, the κ parameters are defined to be "relative infectiousness." It is not clear based on these tables or other information provided by the authors what this infectiousness is "relative" to (especially since $\kappa_P, \kappa_M, \kappa_S$ are set to be one)? I guess what they mean is all those κ parameters are equal to 1, while κ_A is the relative infectiousness of the asymptomatic infections (which is incorrectly written as κ_S in Table 2) is the relative infectiousness parameter for asymptotically infected individuals. This should be clarified.

6. Tables 1-2: The λ terms in these tables are written in "days" but from the equations in (11)-(18), it looks like they should be rates. I assume the authors implemented this correctly but

7. In Page 10 the authors provided some details about the settings used in mif2 but there are more parameters involved in the mif2 runs and I needed to check their code to see these other parameters. The authors should mention, perhaps in the supplementary file, if the parameters that they chose resulted in good convergence diagnostics in mif2 (see for example my next comment.)

8. Lines 97-99: The authors said "we fit parameters to daily deaths for each of the 200 parameter sets using six particle filtering runs with variation in starting values; each run used 100 iterations and 3000 particles." Do they mean "six iterated filtering runs"? And what does "variation in starting values" mean? Does that mean for each parameter set they generated six different initial conditions. Based on their code, it seems like they only allowed for the susceptible class and exposed class to be nonzero at the beginning of their simulation (while the authors mentioned that they fitted the initial exposed class, it was not actually clearly stated in the main text that all others except for the susceptible class were assumed to be zero). Additionally, it is important to explain, perhaps in a supplementary file, if the initial values that mif2 ended up with reasonable values? For example, mif2 might end up with a parameter set suggesting 50% of the population was susceptible and 50% were exposed at the beginning of the simulations, which would not be reasonable. These situations can happen quite often with uncontrolled optimization.

9. The authors' code do not appear to include any constraints for the parameters in Table 2 other than to require them to remain positive during optimization. Did the final values of the parameters resulting from mif2 runs actually remain within the range provided here? If the parameters ended up with values outside of what the authors had in Table 2, this should be mentioned.

10. There were no details about how likelihood was computed at the end and I did not see it in the authors' code either (although I perhaps just missed it). Since mif2 does not provide the correct likelihood (unless all random walks were set to zero at the end), I assume pfilter was used to compute the actual likelihood. But I did not see details on how many particles were used for pfilter. I assume the authors would have estimated the likelihood using pfilter (by running pfilter multiple times then taking the logmeanexp) but this was not mentioned anywhere. Also, the authors should mention the spread of the multiple log likelihood estimates given by pfilter to see if the likelihood estimates are reasonable. They should make sure to attach this in their results if they publish the parameter sets

11. No values of the final parameters were presented in the results and no confidence intervals given. No profiles of the parameter values to confirm that the likelihood has actually been optimized. It is unclear if each of the short time intervals that the models were being fitted to provide enough information to get clearly defined maximum likelihood estimates. The authors should at least provide profiles and confidence intervals resulting from fitting to the first and longest time interval.

12. Line 109: The authors mentioned that R_0 was "estimated" to be β_0 divided by duration of the average infection (as defined by their model structure). But it should be straightforward to just compute the exact R_0 , which is the spectral radius of the next generation matrix generated from the differential equation system that is the basis of the continuous time processes. Then this would involve the various relative infectiousness parameters, as well as the presymptomatic duration. This can affect R_0 significantly.

13. Line 135: How are 50 parameter sets "randomly selected" from the 200 model fits? Are these selections weighted by likelihood? If some parameter sets ended up with higher likelihoods than others, as one might expect from a stochastic fitting algorithm, then parameter sets with higher likelihoods should be weighted more.

14. In Lines 144-146, how were the confidence intervals for R_0 and RE computed given that there are no confidence intervals for the parameters that are used to compute them? Were these just taken from the distribution of values at the final mif2 iteration of each of the 1200 runs? Also, since RE was computed using the "median proportion of the population remaining susceptible," it should be easy to plot the median number of susceptibles generated by the models also and see if these are reasonably consistent with estimates coming from antibody testing, etc.

15. The authors said "We assessed the model forecast performance by computing the mean absolute error of cases and deaths across two weeks after the final date used in the corresponding model fit." This is fine but the authors used Poisson errors when fitting the model, suggesting that they preferred to use a distribution with a variance that scales with the mean. So wouldn't a fairer comparison of the forecast performance would be using Poisson errors instead of absolute errors? Otherwise, why not just use normal errors (equivalent to least squares) from the beginning?

16. The authors should plot the mean number of susceptibles in the population over time to see if the "hidden states" of their model are looking reasonable. In many SEIR models for COVID, the number of susceptibles drop by large, significant fractions, inconsistent with what antibody testing results suggest.

17. I agree with what the authors say in line 29, "most early models quickly became out-of-date as more sophisticated modeling options became available." and line 39 "it is important to understand the value and limitations of early models and to assess their accuracy over time." I suggest that the authors spend more computational effort on fitting these models by generating the profiles and confidence intervals for each parameter that they are fitting and including the MLE point estimates and confidence intervals in their results. It is important to establish confidence first that the model fitting has converged and the fits are reasonable before using this for forecasting and analysis of the effects of different interventions.

Minor comments

1. Lines 60-61: The citations are not in increasing order.

2. Equations (11)-(18): The authors should mention what Δt is before using it. I understand from the text that $\Delta t = 4$ hours, but the symbol should be defined first anyway.

3. Equations (11)-(18): Some of the transition terms here are written as multinomial draws and some as binomial draws. In the multinomial draws, the "transitions" into the same compartment (basically the number that stay in its compartment, such as $d_{E,E}$ for example) is written out explicitly even though this "transition" is not listed in equations (1)-(10). Such transitions are not written out explicitly for the binomial draws. I suggest the authors make a decision on how to treat these "transitions" and use a consistent notation in their equations.

4. Line 126: What does "from the single best fit on April 22 across the 200 parameter sets as defined by negative log likelihood" mean? Do the authors just mean they the parameter set with the best likelihood? I am also still wondering about where how the likelihoods were computed (using pfilter?) and how the best one was found. Weren't they six runs of each of the 200 starting parameter sets (each run having different initial conditions?) so do they mean the parameter set with the best resulting likelihood estimate across the 1200 runs?

5. One important non-pharmaceutical intervention that the authors did not explicitly include in the model is contact tracing and quarantine. This should be mentioned.

Review form: Reviewer 2 (Matthieu Domenech de Cellès)

Recommendation

Major revision is needed (please make suggestions in comments)

Scientific importance: Is the manuscript an original and important contribution to its field?

Acceptable

General interest: Is the paper of sufficient general interest?

Acceptable

Quality of the paper: Is the overall quality of the paper suitable?

Acceptable

Is the length of the paper justified?

Yes

Should the paper be seen by a specialist statistical reviewer?

Yes

Do you have any concerns about statistical analyses in this paper? If so, please specify them explicitly in your report.

Yes

It is a condition of publication that authors make their supporting data, code and materials available - either as supplementary material or hosted in an external repository. Please rate, if applicable, the supporting data on the following criteria.

Is it accessible?

Yes

Is it clear?

N/A

Is it adequate?

N/A

Do you have any ethical concerns with this paper?

No

Comments to the Author

Comments in the pdf document. (See Appendix A)

Review form: Reviewer 3

Recommendation

Major revision is needed (please make suggestions in comments)

Scientific importance: Is the manuscript an original and important contribution to its field?

Good

General interest: Is the paper of sufficient general interest?

Good

Quality of the paper: Is the overall quality of the paper suitable?

Acceptable

Is the length of the paper justified?

Yes

Should the paper be seen by a specialist statistical reviewer?

No

Do you have any concerns about statistical analyses in this paper? If so, please specify them explicitly in your report.

Yes

It is a condition of publication that authors make their supporting data, code and materials available - either as supplementary material or hosted in an external repository. Please rate, if applicable, the supporting data on the following criteria.

Is it accessible?

Yes

Is it clear?

No

Is it adequate?

N/A

Do you have any ethical concerns with this paper?

No

Comments to the Author

In this article, the authors use a mathematical model to analyze the COVID-19 outbreak in Santa Clara county and evaluate its ability to predict future dynamics given limited information about the epidemic. They show that a relatively simple model, which assumes a constant reduction in contact rates during the shelter-in-place period, can accurately predict short-term dynamics and can be still useful in informing policy decisions through scenario analysis. However, as the epidemic unfolds, early models that make simplifying assumptions about the epidemic processes

(e.g., assuming a constant effect of NPIs as done here) may not be able to predict the second wave. The evaluation of the performance of early models is an important topic as mathematical models played critical roles in shaping the responses to the current pandemic. Their analysis is sound, but I have some concerns about the presentation of the work.

The paper lacks discussion of other early models that were used during the current pandemic. Here, the authors consider a stochastic SEIR model that they developed. Given that the main goal of the paper is to "evaluate the accuracy of an early epidemiological compartment model over time to understand the value and limitations of models during unfolding epidemics," I think the authors should provide at least some overview of the earlier COVID-19 models in the introduction and compare them with the model used here. For example, one of the main reasons why the second wave cannot be predicted with the current model is because the transmission rate is modeled as a step function and is not allowed to vary during the forecast. One could imagine allowing the transmission rate to change stochastically over time (e.g., using a random walk prior), which may provide some clue as to when the second wave can occur. Some early models (e.g., [https://doi.org/10.1016/S1473-3099\(20\)30144-4](https://doi.org/10.1016/S1473-3099(20)30144-4)) allowed for such flexibility (although I'm not sure if they were used for forecasting). Other models (e.g., <https://doi.org/10.1038/s41586-020-2405-7>) relied on similar assumptions like this one (assuming constant effects of NPIs) and therefore had difficulty modeling resurgence. I am not at all suggesting that the authors should re-do the analysis, but I think it is important to discuss other modeling approaches (are they broadly similar? or different? do same principles/conclusions still apply?).

If possible, it would be nice if the authors can comment on possible remedies for early models and tradeoffs. For example, new models are almost always needed as we learn more about the disease, but this can be time-consuming. It can also be difficult to compare results from new models with those from earlier models. On the other hand, one could imagine building a very flexible model earlier on, but such a model may have too little power to tell us anything useful. I believe such discussion may make the argument of the paper stronger and can further highlight the importance/limitations of early models.

I was confused about the order in which the results are presented. The authors present the scenario analysis first and then discuss their parameter estimates and model predictions. I think it should be the other way around. Model fits and their accuracy should be shown first (Figures 3 and 4). Estimates of the effective reproduction number should be shown after the model fits to assess the validity of these estimates. The scenario analysis should be presented at the end after the validity of the model fits has been demonstrated.

I had several minor comments about the paper (mostly regarding the writing). Please see below.

Detailed comments:

- Abstract: R_0 is defined as the basic reproduction number but R_E is not defined ("our estimated RE varied"). Maybe better to be consistent and say effective reproduction number (RE)? But not necessary at all...
- Line 9: It would be nice to provide some citations of early models that did this and perhaps briefly discuss them to provide a better overview of early models here.
- Line 14 ("Early models can and should inform policy decisions during an emerging pandemic"): While I agree with the authors, they have not explained why they "should" inform policy decisions at this point. The advantages of early models could be highlighted more explicitly.
- Line 31: I wonder if it's possible to give some sort of references or examples of doing this...
- Line 78: I wonder why Poisson was used instead of the negative binomial?
- Equations: I think it could be useful to show the actual model diagram in the main text rather than in the supplementary file given that the equation is a bit dense. For example, it is not clear from the writing that presymptomatic individuals can be either mildly or severely symptomatic. It's also not entirely trivial to go back and forth between equations 1-10, equations 11-18, and tables 1-2...

- Equation 11: It's missing a time-varying transmission rate parameter. It only has relative infectiousness right now...
- Equations 12-16: I believe $dE, E, dI_p, I_p,$ and dI_s, I_s terms are not needed in the equations (because the authors are drawing a multinomial from E to I_a and I_p); these terms also don't appear in equations 1-10. The current formulation suggests that the authors are drawing three quantities from E (going from E to E, E to $I_p,$ and E to I_m). I suspect this is just a notation problem, and not an implementation problem though...
- Table 1-2: Table 2 caption reads "Parameter range estimates that are not location-specific" which could imply that Table 1 estimates are location specific? I can see that some parameters, such as time from symptom onset to hospitalization, are location-specific but others, such as incubation periods, are probably not. Also, table 2 should say κ_A rather than κ_S . I also wonder why point estimates were used for some and not for others. I suspect this is due to the location specificity of some parameters, but I think it needs to be explained more clearly.
- Line 92: I could not understand how the transmission rate is modeled here. It's briefly explained in Lines 109-112, which reads as if different σ values are estimated for each week. I could not figure this out until I got to Line 225 ("especially when assuming a constant impact of non-pharmaceutical interventions, as this model does"). The duration of shelter-in-place should also be noted at this point.
- Line 119: I suspect the results can vary depending on how much social distancing is relaxed vs how much transmission rates for symptomatic individuals are reduced. This should be explained more clearly.
- Line 127: I think it would be informative to provide what summary statistics you're considering at this point.
- Line 133: It should be noted that uncertainties associated with β_0 and σ are also not taken into account. I understand that there are uncertainties in β_0 and σ due to 200 parameter sets that the authors consider but each fit has its own parameter uncertainty. It would be also nice to cite Elder et al. 2006 here (<https://doi.org/10.1073/pnas.0600816103>).
- Line 139: It should be noted that this does not take into account the delay between symptom onset and reporting of cases (and so the predicted cases can be systematically biased).
- Figure 1: Are these CIs also based on 200 parameter sets (as opposed to something like profiling)?
- Figure 1: It might be useful to show the daily number of infections for the readers to assess estimates of R_e .
- Line 148: Is this based on any random "one parameter set" or the best fit?
- Line 151: What is meant by the number of concurrent infections? Does it include everyone who has not recovered including the hospitalized individuals?
- Figure 2: It seems like predictions are made by simulating epidemics beginning from the initial time without conditioning on the data. Instead, predictions should be made by conditioning on the data first, and then simulating forward. The first step can be done by using a pfilter function and taking the particle filtered trajectories. This will remove simulations that do not match the data (e.g., some simulations reach <10 cumulative deaths as of June 1st) and reduce uncertainty accordingly.
- Line 154: "removing an additional 80% and 70% of infective contacts from severe and mild infections respectively" infective \rightarrow infective?
- Line 183: I'm a bit confused by how R_E s are compared here. Since a constant reduction in transmission rate is assumed throughout the shelter-in-place period, the model fit based on the data until July estimates that R_E for earlier dates (e.g., early April) is also around 0.8. On the other hand, earlier fits estimate that R_E in April is much lower. It would be nice for the authors to comment on this discrepancy and how one should go about interpreting such estimates. For example, when the model is fitted to data until July, are R_E s for earlier weeks re-estimated as well? Or are previous R_E estimates accounted for in the new fits somehow?
- Line 183: "Following a spike in deaths, R_E increased to 0.977 on April 22" Shouldn't R_E increase before a spike in deaths because R_E s reflect infections?
- Line 184: Again, forecasts should rely on filtered trajectories, rather than based on unconditional simulations.
- Figure 4 caption: "Mean absolute error for Colors indicate the percentage of errors that are over-

estimates, where red indicates a high percentage of overestimates and blue indicates a high percentage of under-estimates" Typo at the beginning of this sentence ("Mean absolute error for")?

- Line 284-288: I was unsure whether this discussion was about the model presented in the paper or about other early models in general.

- I wonder if the code provided is the correct code? It seems like multiple boxes were used for each infection class and mobility data were also included somehow, but I don't see any descriptions in the paper...

Decision letter (RSPB-2020-2281.R0)

24-Nov-2020

Dear Dr Mordecai:

I am writing to inform you that your manuscript RSPB-2020-2281 entitled "The impact of long-term non-pharmaceutical interventions on COVID-19 epidemic dynamics and control: the value and limitations of early models" has, in its current form, been rejected for publication in Proceedings B.

This action has been taken on the advice of referees, who have recommended that substantial revisions are necessary. With this in mind we would be happy to consider a resubmission, provided the comments of the referees are fully addressed. However please note that this is not a provisional acceptance.

Sincerely,

Professor Hans Heesterbeek

Associate Editor

Board Member: 1

Comments to Author:

This is an interesting manuscript that examines how well a specific compartment model performs in projection COVID-19 dynamics. There are three referee reports and they too found the manuscript interesting. That said, they raised a large number of concerns that I think would need to be addressed before one can adequately judge the suitability of the manuscript for publication.

The referees are extremely thorough and I think attention needs to be given to all of their comments, but a few of them seem particularly important:

First, two referees raised concerns about the lack of information on whether the numerical procedure has converged when fitting the model. This is a very important issue and I think it needs to be thoroughly addressed.

Second, R2 wondered if the model might be mis specified in some way since certain aspects of the fit do not seem to improve as more data is included.

Third, there is also a general request that more information be given about the details of the model fitting and implementation, to the point where it would be possible for a reader to reconstruct the results if desired.

Finally, I agree with R3 that it would be good to include a more complete discussion of other models that address the same issue. I realize that the authors cannot possibly include any formal analysis of other models in a paper of this length but if the goal is to examine how well mathematical models of epidemics perform in the early stages of an outbreak, then I think other models should at least be discussed in as much detail as possible.

Reviewer(s)' Comments to Author:

Referee: 1

Comments to the Author(s)

The authors present a mathematical model of COVID-19 that accounts for non-pharmaceutical interventions. They used iterated filtering in the pomp R package to fit a stochastic discrete time version of their model. They evaluated the accuracy of the predictions of the model over time "to understand the value and limitations of models during unfolding epidemics."

The paper is well-written and interesting. The goal of the paper is very good. However, I have some concerns about the convergence of the parameter fitting using mif2. This convergence issue is always an important one when using stochastic fitting algorithms and it was not clearly shown by the authors that their parameters had converged. Here are some more specific comments:

Comments for Major Revision

1. Equations (1)-(10) are written in differential equation form where the transition terms $d_{X,Y}$ are transition rates. But the terms $d_{X,Y}$ as defined in equations (11)-(18) are defined for discrete time models. I understand that the continuous time stochastic process was simulated using Euler approximation. However the equations are inconsistent. There are not equations for the continuous time stochastic process. Equations (1)-(10) are written as if they are deterministic differential equations and Equations (11)-(18) are equations for discrete-time stochastic process. The authors need to choose a more consistent way to present their model.

2. Given the inconsistencies of Equation (1)-(10) and (11)-(18), I think it would be better to just put the diagram in the supplementary file in the main text and remove the differential equation system in (1)-(10) which the authors say they did not use.

3. The authors explain that they will use a transmission "parameter" β that will change over time. However this β does not appear in equations (1)-(10) or (11)-(18). I assume it is supposed to be part of the expression in equation (11) and was just accidentally dropped from there? This would be the normal location for it and what their online code suggests.

4. Regarding the parameter values chosen in Table 1. It is important to note that parameters estimated from other sources of data do not often translate to the same way in a population-level model. While I am not suggesting that the authors need to change their parameters, they should note that this can be an important issue. For example when using $\lambda_A=1/5$ days that the authors chose for the asymptomatic rate (I should note that the authors actually wrote in Table 1 that $\lambda_A=5$ days which is inconsistent with equation (13) which clearly shows that λ_A should be a rate and not a period). Since the authors say in lines 74-75 that "We use an Euler approximation of the continuous time process," the binomial draws involving λ_A in equation (13) is supposed to approximate a continuous time process which means λ_A is supposed to be an exponential rate. However, I expect that most observations of the asymptomatic period does not have exponential distribution, but a more "central" distribution, such as an Erlang distribution.

5. In Tables 1-2, the κ parameters are defined to be "relative infectiousness." It is not clear based on these tables or other information provided by the authors what this infectiousness is "relative" to (especially since $\kappa_P, \kappa_M, \kappa_S$ are set to be one)? I guess what they mean is all those κ parameters are equal to 1, while κ_A is the relative infectiousness of the asymptomatic infections (which is incorrectly written as κ_S in Table 2) is the relative infectiousness parameter for asymptotically infected individuals. This should be clarified.

6. Tables 1-2: The λ terms in these tables are written in "days" but from the equations in (11)-(18), it looks like they should be rates. I assume the authors implemented this correctly but

7. In Page 10 the authors provided some details about the settings used in mif2 but there are more parameters involved in the mif2 runs and I needed to check their code to see these other parameters. The authors should mention, perhaps in the supplementary file, if the parameters that they chose resulted in good convergence diagnostics in mif2 (see for example my next comment.)

8. Lines 97-99: The authors said "we fit parameters to daily deaths for each of the 200 parameter sets using six particle filtering runs with variation in starting values; each run used 100 iterations and 3000 particles." Do they mean "six iterated filtering runs"? And what does "variation in starting values" mean? Does that mean for each parameter set they generated six different initial conditions. Based on their code, it seems like they only allowed for the susceptible class and exposed class to be nonzero at the beginning of their simulation (while the authors mentioned that they fitted the initial exposed class, it was not actually clearly stated in the main text that all others except for the susceptible class were assumed to be zero). Additionally, it is important to explain, perhaps in a supplementary file, if the initial values that mif2 ended up with reasonable values? For example, mif2 might end up with a parameter set suggesting 50% of the population was susceptible and 50% were exposed at the beginning of the simulations, which would not be reasonable. These situations can happen quite often with uncontrolled optimization.

9. The authors' code do not appear to include any constraints for the parameters in Table 2 other than to require them to remain positive during optimization. Did the final values of the parameters resulting from mif2 runs actually remain within the range provided here? If the parameters ended up with values outside of what the authors had in Table 2, this should be mentioned.

10. There were no details about how likelihood was computed at the end and I did not see it in the authors' code either (although I perhaps just missed it). Since mif2 does not provide the

correct likelihood (unless all random walks were set to zero at the end), I assume pfilter was used to compute the actual likelihood. But I did not see details on how many particles were used for pfilter. I assume the authors would have estimated the likelihood using pfilter (by running pfilter multiple times then taking the logmeanexp) but this was not mentioned anywhere. Also, the authors should mention the spread of the multiple log likelihood estimates given by pfilter to see if the likelihood estimates are reasonable. They should make sure to attach this in their results if they publish the parameter sets

11. No values of the final parameters were presented in the results and no confidence intervals given. No profiles of the parameter values to confirm that the likelihood has actually been optimized. It is unclear if each of the short time intervals that the models were being fitted to provide enough information to get clearly defined maximum likelihood estimates. The authors should at least provide profiles and confidence intervals resulting from fitting to the first and longest time interval.

12. Line 109: The authors mentioned that R_0 was "estimated" to be β_0 divided by duration of the average infection (as defined by their model structure). But it should be straightforward to just compute the exact R_0 , which is the spectral radius of the next generation matrix generated from the differential equation system that is the basis of the continuous time processes. Then this would involve the various relative infectiousness parameters, as well as the presymptomatic duration. This can affect R_0 significantly.

13. Line 135: How are 50 parameter sets "randomly selected" from the 200 model fits? Are these selections weighted by likelihood? If some parameter sets ended up with higher likelihoods than others, as one might expect from a stochastic fitting algorithm, then parameter sets with higher likelihoods should be weighted more.

14. In Lines 144-146, how were the confidence intervals for R_0 and RE computed given that there are no confidence intervals for the parameters that are used to compute them? Were these just taken from the distribution of values at the final mif2 iteration of each of the 1200 runs? Also, since RE was computed using the "median proportion of the population remaining susceptible," it should be easy to plot the median number of susceptibles generated by the models also and see if these are reasonably consistent with estimates coming from antibody testing, etc.

15. The authors said "We assessed the model forecast performance by computing the mean absolute error of cases and deaths across two weeks after the final date used in the corresponding model fit." This is fine but the authors used Poisson errors when fitting the model, suggesting that they preferred to use a distribution with a variance that scales with the mean. So wouldn't a fairer comparison of the forecast performance would be using Poisson errors instead of absolute errors? Otherwise, why not just use normal errors (equivalent to least squares) from the beginning?

16. The authors should plot the mean number of susceptibles in the population over time to see if the "hidden states" of their model are looking reasonable. In many SEIR models for COVID, the number of susceptibles drop by large, significant fractions, inconsistent with what antibody testing results suggest.

17. I agree with what the authors say in line 29, "most early models quickly became out-of-date as more sophisticated modeling options became available." and line 39 "it is important to understand the value and limitations of early models and to assess their accuracy over time." I suggest that the authors spend more computational effort on fitting these models by generating the profiles and confidence intervals for each parameter that they are fitting and including the MLE point estimates and confidence intervals in their results. It is important to establish confidence first that the model fitting has converged and the fits are reasonable before using this for forecasting and analysis of the effects of different interventions.

Minor comments

1. Lines 60-61: The citations are not in increasing order.
2. Equations (11)-(18): The authors should mention what $\$dt\$$ is before using it. I understand from the text that $\$dt = 4\$$ hours, but the symbol should be defined first anyway.
3. Equations (11)-(18): Some of the transition terms here are written as multinomial draws and some as binomial draws. In the multinomial draws, the "transitions" into the same compartment (basically the number that stay in its compartment, such as $\$d_{\{E,E\}}\$$ for example) is written out explicitly even though this "transition" is not listed in equations (1)-(10). Such transitions are not written out explicitly for the binomial draws. I suggest the authors make a decision on how to treat these "transitions" and use a consistent notation in their equations.
4. Line 126: What does "from the single best fit on April 22 across the 200 parameter sets as defined by negative log likelihood" mean? Do the authors just mean they the parameter set with the best likelihood? I am also still wondering about where how the likelihoods were computed (using pfilter?) and how the best one was found. Weren't they six runs of each of the 200 starting parameter sets (each run having different initial conditions?) so do they mean the parameter set with the best resulting likelihood estimate across the 1200 runs?
5. One important non-pharmaceutical intervention that the authors did not explicitly include in the model is contact tracing and quarantine. This should be mentioned.

Referee: 2

Comments to the Author(s)

Comments in the pdf document.

Referee: 3

Comments to the Author(s)

In this article, the authors use a mathematical model to analyze the COVID-19 outbreak in Santa Clara county and evaluate its ability to predict future dynamics given limited information about the epidemic. They show that a relatively simple model, which assumes a constant reduction in contact rates during the shelter-in-place period, can accurately predict short-term dynamics and can be still useful in informing policy decisions through scenario analysis. However, as the epidemic unfolds, early models that make simplifying assumptions about the epidemic processes (e.g., assuming a constant effect of NPIs as done here) may not be able to predict the second wave. The evaluation of the performance of early models is an important topic as mathematical models played critical roles in shaping the responses to the current pandemic. Their analysis is sound, but I have some concerns about the presentation of the work.

The paper lacks discussion of other early models that were used during the current pandemic. Here, the authors consider a stochastic SEIR model that they developed. Given that the main goal of the paper is to "evaluate the accuracy of an early epidemiological compartment model over time to understand the value and limitations of models during unfolding epidemics," I think the authors should provide at least some overview of the earlier COVID-19 models in the introduction and compare them with the model used here. For example, one of the main reasons why the second wave cannot be predicted with the current model is because the transmission rate is modeled as a step function and is not allowed to vary during the forecast. One could imagine allowing the transmission rate to change stochastically over time (e.g., using a random walk prior), which may provide some clue as to when the second wave can occur. Some early models (e.g., [https://doi.org/10.1016/S1473-3099\(20\)30144-4](https://doi.org/10.1016/S1473-3099(20)30144-4)) allowed for such flexibility (although I'm not sure if they were used for forecasting). Other models (e.g., <https://doi.org/10.1038/s41586-020-2405-7>) relied on similar assumptions like this one (assuming constant effects of NPIs) and therefore had difficulty modeling resurgence. I am not at all suggesting that the authors should

re-do the analysis, but I think it is important to discuss other modeling approaches (are they broadly similar? or different? do same principles/conclusions still apply?).

If possible, it would be nice if the authors can comment on possible remedies for early models and tradeoffs. For example, new models are almost always needed as we learn more about the disease, but this can be time-consuming. It can also be difficult to compare results from new models with those from earlier models. On the other hand, one could imagine building a very flexible model earlier on, but such a model may have too little power to tell us anything useful. I believe such discussion may make the argument of the paper stronger and can further highlight the importance/limitations of early models.

I was confused about the order in which the results are presented. The authors present the scenario analysis first and then discuss their parameter estimates and model predictions. I think it should be the other way around. Model fits and their accuracy should be shown first (Figures 3 and 4). Estimates of the effective reproduction number should be shown after the model fits to assess the validity of these estimates. The scenario analysis should be presented at the end after the validity of the model fits has been demonstrated.

I had several minor comments about the paper (mostly regarding the writing). Please see below.

Detailed comments:

- Abstract: R_0 is defined as the basic reproduction number but R_E is not defined ("our estimated RE varied"). Maybe better to be consistent and say effective reproduction number (RE)? But not necessary at all...

- Line 9: It would be nice to provide some citations of early models that did this and perhaps briefly discuss them to provide a better overview of early models here.

- Line 14 ("Early models can and should inform policy decisions during an emerging pandemic"): While I agree with the authors, they have not explained why they "should" inform policy decisions at this point. The advantages of early models could be highlighted more explicitly.

- Line 31: I wonder if it's possible to give some sort of references or examples of doing this...

- Line 78: I wonder why Poisson was used instead of the negative binomial?

- Equations: I think it could be useful to show the actual model diagram in the main text rather than in the supplementary file given that the equation is a bit dense. For example, it is not clear from the writing that presymptomatic individuals can be either mildly or severely symptomatic. It's also not entirely trivial to go back and forth between equations 1-10, equations 11-18, and tables 1-2...

- Equation 11: It's missing a time-varying transmission rate parameter. It only has relative infectiousness right now...

- Equations 12-16: I believe $dE, E, dI_p, I_p,$ and dI_s, I_s terms are not needed in the equations (because the authors are drawing a multinomial from E to I_a and I_p); these terms also don't appear in equations 1-10. The current formulation suggests that the authors are drawing three quantities from E (going from E to E, E to $I_p,$ and E to I_m). I suspect this is just a notation problem, and not an implementation problem though...

- Table 1-2: Table 2 caption reads "Parameter range estimates that are not location-specific" which could imply that Table 1 estimates are location specific? I can see that some parameters, such as time from symptom onset to hospitalization, are location-specific but others, such as incubation periods, are probably not. Also, table 2 should say κ_A rather than κ_S . I also wonder why point estimates were used for some and not for others. I suspect this is due to the location specificity of some parameters, but I think it needs to be explained more clearly.

- Line 92: I could not understand how the transmission rate is modeled here. It's briefly explained in Lines 109-112, which reads as if different σ values are estimated for each week. I could not figure this out until I got to Line 225 ("especially when assuming a constant impact of non-pharmaceutical interventions, as this model does"). The duration of shelter-in-place should also be noted at this point.

- Line 119: I suspect the results can vary depending on how much social distancing is relaxed vs how much transmission rates for symptomatic individuals are reduced. This should be explained more clearly.
- Line 127: I think it would be informative to provide what summary statistics you're considering at this point.
- Line 133: It should be noted that uncertainties associated with β_0 and σ are also not taken into account. I understand that there are uncertainties in β_0 and σ due to 200 parameter sets that the authors consider but each fit has its own parameter uncertainty. It would be also nice to cite Elder et al. 2006 here (<https://doi.org/10.1073/pnas.0600816103>).
- Line 139: It should be noted that this does not take into account the delay between symptom onset and reporting of cases (and so the predicted cases can be systematically biased).
- Figure 1: Are these CIs also based on 200 parameter sets (as opposed to something like profiling)?
- Figure 1: It might be useful to show the daily number of infections for the readers to assess estimates of R_e .
- Line 148: Is this based on any random "one parameter set" or the best fit?
- Line 151: What is meant by the number of concurrent infections? Does it include everyone who has not recovered including the hospitalized individuals?
- Figure 2: It seems like predictions are made by simulating epidemics beginning from the initial time without conditioning on the data. Instead, predictions should be made by conditioning on the data first, and then simulating forward. The first step can be done by using a pfilter function and taking the particle filtered trajectories. This will remove simulations that do not match the data (e.g., some simulations reach <10 cumulative deaths as of June 1st) and reduce uncertainty accordingly.
- Line 154: "removing an additional 80% and 70% of infective contacts from severe and mild infections respectively" infective \rightarrow infective?
- Line 183: I'm a bit confused by how R_E s are compared here. Since a constant reduction in transmission rate is assumed throughout the shelter-in-place period, the model fit based on the data until July estimates that R_E for earlier dates (e.g., early April) is also around 0.8. On the other hand, earlier fits estimate that R_E in April is much lower. It would be nice for the authors to comment on this discrepancy and how one should go about interpreting such estimates. For example, when the model is fitted to data until July, are R_E s for earlier weeks re-estimated as well? Or are previous R_E estimates accounted for in the new fits somehow?
- Line 183: "Following a spike in deaths, R_E increased to 0.977 on April 22" Shouldn't R_E increase before a spike in deaths because R_E s reflect infections?
- Line 184: Again, forecasts should rely on filtered trajectories, rather than based on unconditional simulations.
- Figure 4 caption: "Mean absolute error for Colors indicate the percentage of errors that are over-estimates, where red indicates a high percentage of overestimates and blue indicates a high percentage of under-estimates" Typo at the beginning of this sentence ("Mean absolute error for")?
- Line 284-288: I was unsure whether this discussion was about the model presented in the paper or about other early models in general.
- I wonder if the code provided is the correct code? It seems like multiple boxes were used for each infection class and mobility data were also included somehow, but I don't see any descriptions in the paper...

Author's Response to Decision Letter for (RSPB-2020-2281.R0)

See Appendix B.

RSPB-2021-0811.R0

Review form: Reviewer 1

Recommendation

Major revision is needed (please make suggestions in comments)

Scientific importance: Is the manuscript an original and important contribution to its field?

Acceptable

General interest: Is the paper of sufficient general interest?

Acceptable

Quality of the paper: Is the overall quality of the paper suitable?

Acceptable

Is the length of the paper justified?

Yes

Should the paper be seen by a specialist statistical reviewer?

No

Do you have any concerns about statistical analyses in this paper? If so, please specify them explicitly in your report.

Yes

It is a condition of publication that authors make their supporting data, code and materials available - either as supplementary material or hosted in an external repository. Please rate, if applicable, the supporting data on the following criteria.

Is it accessible?

Yes

Is it clear?

Yes

Is it adequate?

Yes

Do you have any ethical concerns with this paper?

No

Comments to the Author

The authors presented a simple mathematical model of COVID-19, implemented as a stochastic discrete-time model and fit to data using maximization via iterated filtering using the R package pomp. They then evaluated the performance and accuracy of the predictions of the model.

The paper is well-written and the revised version is a big improvement over the authors' initial submission. Many of the original issues with their original manuscript (especially with regard to description of the models and methods) have been resolved. However, I still have some concerns about the convergence of the parameter fitting using mif2. Here are some more specific comments:

1. The most important issue for me is still the lack of parameter profiles. In Figure S6 the authors present the parameter sets versus the likelihood. These could be considered "poor man's profiles." Unfortunately since these are not properly computed profiles, we cannot really use them to find confidence intervals of the parameter. More worryingly, these plots seem to indicate that most parameters do not yet have a properly computed maximum likelihood estimate. Actually running proper profiles may help the authors actually maximize their parameter sets. Profiling is often required to get good maximization which one often cannot get from just running a large number of MIF iterations (since MIF iterations can often get stuck or have small variance after the first few MIF runs, especially with the small number of particles that the authors used, which is often not enough when fitting real data). Big improvements in likelihood can be possible after running profiles across all fitted parameters, so I strongly suggest the authors at least try to run proper profiles and present them in their manuscript. I should note that unfortunately sometimes proper profiling ends up giving us flat profiles, even when the initial MIF runs yield some curvature. This is important to learn, even if the results end up just showing us that the parameters are very much unidentifiable (usually because the model really does not fit).

2. After doing the profiles, the authors can then use the proper 95% confidence intervals from the profiles instead of using "two log likelihood units" which is a bit arbitrary.

3. If the authors are able to get sharper MLEs for their parameters, the range of susceptible values shown in Figures S8 may change (although unfortunately sometimes these values end up being less realistic). They may also be able to determine which of the two very distinct trajectories in the first panel is a higher likelihood scenario. Otherwise this gives a huge range of estimates on the total number of people that have been infected at any time point. While the authors point out that "majority of parameter sets within for all fits suggested a large percent of the population remained susceptible on the fit date," this is not enough. Because sometimes a majority of MIF runs might end up in the same parameter region, not because that is actually the highest likelihood region but just because of how the gradient of the likelihood surface is near that local maximum. So I would not use the density of the parameter values at the end of the runs as indication of what is "most likely" (also an issue for other figures).

4. I don't think Google Drive is the best place to make results available. Couldn't these have been also uploaded to GitHub? Or use DataDryad?

Review form: Reviewer 2 (Matthieu Domenech de Cellès)

Recommendation

Accept with minor revision (please list in comments)

Scientific importance: Is the manuscript an original and important contribution to its field?

Good

General interest: Is the paper of sufficient general interest?

Good

Quality of the paper: Is the overall quality of the paper suitable?

Good

Is the length of the paper justified?

No

Should the paper be seen by a specialist statistical reviewer?

No

Do you have any concerns about statistical analyses in this paper? If so, please specify them explicitly in your report.

No

It is a condition of publication that authors make their supporting data, code and materials available - either as supplementary material or hosted in an external repository. Please rate, if applicable, the supporting data on the following criteria.

Is it accessible?

Yes

Is it clear?

Yes

Is it adequate?

Yes

Do you have any ethical concerns with this paper?

No

Comments to the Author

Comments in attached pdf. (See Appendix C)

Review form: Reviewer 3

Recommendation

Accept with minor revision (please list in comments)

Scientific importance: Is the manuscript an original and important contribution to its field?

Good

General interest: Is the paper of sufficient general interest?

Good

Quality of the paper: Is the overall quality of the paper suitable?

Good

Is the length of the paper justified?

Yes

Should the paper be seen by a specialist statistical reviewer?

No

Do you have any concerns about statistical analyses in this paper? If so, please specify them explicitly in your report.

Yes

It is a condition of publication that authors make their supporting data, code and materials available - either as supplementary material or hosted in an external repository. Please rate, if applicable, the supporting data on the following criteria.

Is it accessible?

Yes

Is it clear?

Yes

Is it adequate?

Yes

Do you have any ethical concerns with this paper?

No

Comments to the Author

In this article, the authors evaluate the performance of early models in predicting future epidemic dynamics and evaluating intervention measures. Previously, I suggested that the paper lacks discussion of other studies. The authors have addressed my concern---they provide a reasonably thorough review of limitations and advantages of early-outbreak models in the context of the current pandemic as well as previous outbreaks. I think this paper provides important and relevant insights into modeling practices. I have three major comments and several minor comments (mostly about the writing).

Major comments:

- In table 1, the authors summarize possible modeling options and their tradeoffs. These modeling decisions are discussed throughout the methods section, but the table itself is rarely discussed. I wonder if the authors can try to highlight the table more explicitly (perhaps in the discussion) so that other modelers can refer to the table in making modeling decisions.

- The discussion section primarily focuses on the limitations and values of early models. But I wonder if there is anything that the early models could have done better at the beginning of the current pandemic (even with limited information). For example, is there anything that the authors would have done differently in hindsight? Or were there assumptions that other early models made but turned out to be wrong or inaccurate? It seems like it would be important to include a brief discussion on ways to improve early models---and such discussion will be informative for early models of future outbreaks.

- The added supplementary figures illustrate convergence to some degree. For example, R_0 estimates (Figure S6) show a clear likelihood curvature, demonstrating that parameters are well identified. Parameters that are sampled, rather than estimated, give flat likelihood curves, meaning that the data provide little information about those parameters, suggesting that they are difficult to estimate (as explained by the authors in the manuscript; and this is OK). However, I find parameter estimates for E_0 a bit weird. It seems that log-likelihood values increase as E_0 increases, but parameter estimates rarely go over $E_0 \approx 6$. I would expect log-likelihood values to peak eventually and decrease as E_0 continues to increase. I wonder if this has to do with the fact that E_0 was sampled between 0 and 6, but the mif2 algorithm was run using either an insufficient number of iterations or a small standard deviation (not allowing the parameters to deviate much from their starting values). This might be a relatively minor issue given that E_0 is not a focal parameter but it is still concerning as it suggests that fits may not have converged. For example, R_0 may look like it converged at a given E_0 value (e.g., at some likelihood slice), but if E_0 has not converged yet, then other parameter estimates can be suboptimal. Also, please clarify how relative log-likelihoods are defined.

Minor comments:

L25: "Early models must be built and calibrated to data rapidly with highly incomplete and uncertain data" redundant phrasing?

L126: "We divided the population into states with respect to SARS-CoV-2" do you mean "with respect to SARS-CoV-2 infection"?

L131: "We assumed that transitions between states were simulated as binomial or multinomial processes, which treat periods within each state as being exponentially distributed" Does this not correspond to geometric distribution, rather than exponential, given that it's a discrete-time model?

L151: "We include σ_{WFH} and the 152 work-from-home start date as two of the sampled parameters (see Table 3) but allow σ_{SIP} to be estimated by the model (see "Fitting the Model" below)." It should be justified why σ_{WFH} was sampled, instead of being estimated.

L185: "This median is between ..." I understand this sentence, but it is awkward and confusing (especially the "and used in" part). Please rewrite.

L205: "scaling by the estimated β_0 by σ_{SIP} and the median proportion of the population" => "scaling the estimated β_0 by σ_{SIP} and ..."?

L206: Why 200 simulations here instead of 1200?

L220--223: Shouldn't the number of epidemic forecasts correspond to 625 times the number of parameter sets?

L229: "but narrow because we ignored uncertainty in the parameters listed in Table 1, and thus should be interpreted with caution" It is also narrow because uncertainty in estimated parameters for each fit is not taken into account (though, in practice, these uncertainties might be all washed away by simulating across multiple parameter sets and may not matter too much).

L230: This should be table 2.

L234--239: Please divide this sentence up---it is difficult to understand. It is also not at all clear what the scoring is trying to measure. For example, why do you only consider values that are exactly equal to the observed values? Why does F suddenly appear? Why does the first summation not have any indices?

L241: It's not clear which quantity is being compared with cases at this point in the text. Please clarify.

L261: "Counterfactuals" This section needs to be explained more clearly and requires a better preface. It is confusing what the purpose of this section is when the authors suddenly talk about diverging counterfactual scenarios (even for a reviewer who has already read through the paper once). It's also not exactly clear what the third counterfactual is trying to achieve (presumably to show that intervention introduced later will require additional measures to match the effectiveness of the intervention introduced earlier?).

L294--308: I wonder if it's possible for the authors to add RE values in Figure 4 (maybe in the top left corner of each panel) instead of referring to Figure 3. It's difficult to go back and forth two figures to try to figure out the RE values that correspond to the scenarios presented in Figure 4, especially given that Figure 4 is presented at a 3 week interval, whereas Figure 3 is presented at a 1 week interval.

L306--308: I don't see any underestimation...Where is this result presented?

L309: In reality, there are presumably considerable lags between when people develop symptoms and when cases are reported (and there were probably under-reporting of symptomatic cases as

well). So I'm surprised that the number of symptomatic infections fits well with reported cases. Any comments from the authors? This is very briefly mentioned in L395 ("lagged one week for reporting lag") but I think it should be explained earlier (in Model assessment section).

L320: "We originally designed and fit our model" => "designed and fitted"?

L407: "This pattern occurred throughout the U.S. during the summer resurgence" any citations?

Table 1: "More states increases the number of parameters and will be harder to fit" increase => increase?

Table 1: "Exponentially distributed rates vs Erlang distributed rates" This sounds like rates are distributed. Please clarify.

Figure 1: "SafeGraph" is never mentioned in the text. Maybe remove unnecessary details to avoid confusion?

Figure S6: axis labels overlap

Decision letter (RSPB-2021-0811.R0)

21-May-2021

Dear Dr Mordecai:

Your manuscript has now been peer reviewed and the reviews have been assessed by an Associate Editor. The reviewers' comments (not including confidential comments to the Editor) and the comments from the Associate Editor are included at the end of this email for your reference. As you will see, the reviewers and the Editors have raised some concerns with your manuscript and we would like to invite you to revise your manuscript to address them.

Research ethics:

Use of animals and field studies:

It is a condition of publication that you make available the data and research materials supporting the results in the article (<https://royalsociety.org/journals/authors/author-guidelines/#data>). Datasets should be deposited in an appropriate publicly available repository and details of the associated accession number, link or DOI to the datasets must be included in the Data Accessibility section of the article (<https://royalsociety.org/journals/ethics-policies/data-sharing-mining/>). Reference(s) to datasets should also be included in the reference list of the article with DOIs (where available).

If you wish to submit your data to Dryad (<http://datadryad.org/>) and have not already done so you can submit your data via this link [http://datadryad.org/submit?journalID=RSPB&manu=\(Document not available\)](http://datadryad.org/submit?journalID=RSPB&manu=(Document%20not%20available)), which will take you to your unique entry in the Dryad repository.

Please submit a copy of your revised paper within three weeks. If we do not hear from you within this time your manuscript will be rejected. If you are unable to meet this deadline please let us know as soon as possible, as we may be able to grant a short extension.

Best wishes,
Professor Hans Heesterbeek
mailto: proceedingsb@royalsociety.org

Associate Editor

Comments to Author:

The authors have done an excellent job of revising the manuscript. I am nevertheless recommending another revision for two reasons. First, one of the referees makes a few more very helpful suggestions that will increase the impact of the paper. Second, another referee is still not satisfied with some aspects of the statistical analysis. Unfortunately, this is far enough outside my own area of expertise that I cannot speak to the issue myself and so I think it is important to have the authors address this issue in some way.

Reviewer(s)' Comments to Author:

Referee: 2

Comments to the Author(s).
Comments in attached pdf.

Referee: 3

Comments to the Author(s).

In this article, the authors evaluate the performance of early models in predicting future epidemic dynamics and evaluating intervention measures. Previously, I suggested that the paper lacks discussion of other studies. The authors have addressed my concern---they provide a reasonably thorough review of limitations and advantages of early-outbreak models in the context of the current pandemic as well as previous outbreaks. I think this paper provides important and relevant insights into modeling practices. I have three major comments and several minor comments (mostly about the writing).

Major comments:

- In table 1, the authors summarize possible modeling options and their tradeoffs. These modeling decisions are discussed throughout the methods section, but the table itself is rarely discussed. I wonder if the authors can try to highlight the table more explicitly (perhaps in the discussion) so that other modelers can refer to the table in making modeling decisions.

- The discussion section primarily focuses on the limitations and values of early models. But I wonder if there is anything that the early models could have done better at the beginning of the current pandemic (even with limited information). For example, is there anything that the authors would have done differently in hindsight? Or were there assumptions that other early models made but turned out to be wrong or inaccurate? It seems like it would be important to include a brief discussion on ways to improve early models---and such discussion will be informative for early models of future outbreaks.

- The added supplementary figures illustrate convergence to some degree. For example, R_0 estimates (Figure S6) show a clear likelihood curvature, demonstrating that parameters are well identified. Parameters that are sampled, rather than estimated, give flat likelihood curves, meaning that the data provide little information about those parameters, suggesting that they are difficult to estimate (as explained by the authors in the manuscript; and this is OK). However, I find parameter estimates for E_0 a bit weird. It seems that log-likelihood values increase as E_0 increases, but parameter estimates rarely go over $E_0 > 6$. I would expect log-likelihood values to peak eventually and decrease as E_0 continues to increase. I wonder if this has to do with the fact

that E_0 was sampled between 0 and 6, but the mif2 algorithm was run using either an insufficient number of iterations or a small standard deviation (not allowing the parameters to deviate much from their starting values). This might be a relatively minor issue given that E_0 is not a focal parameter but it is still concerning as it suggests that fits may not have converged. For example, R_0 may look like it converged at a given E_0 value (e.g., at some likelihood slice), but if E_0 has not converged yet, then other parameter estimates can be suboptimal. Also, please clarify how relative log-likelihoods are defined.

Minor comments:

L25: "Early models must be built and calibrated to data rapidly with highly incomplete and uncertain data" redundant phrasing?

L126: "We divided the population into states with respect to SARS-CoV-2" do you mean "with respect to SARS-CoV-2 infection"?

L131: "We assumed that transitions between states were simulated as binomial or multinomial processes, which treat periods within each state as being exponentially distributed" Does this not correspond to geometric distribution, rather than exponential, given that it's a discrete-time model?

L151: "We include σ_{WFH} and the 152 work-from-home start date as two of the sampled parameters (see Table 3) but allow σ_{SIP} to be estimated by the model (see "Fitting the Model" below)." It should be justified why σ_{WFH} was sampled, instead of being estimated.

L185: "This median is between ..." I understand this sentence, but it is awkward and confusing (especially the "and used in" part). Please rewrite.

L205: "scaling by the estimated β_0 by σ_{SIP} and the median proportion of the population" => "scaling the estimated β_0 by σ_{SIP} and ..."?

L206: Why 200 simulations here instead of 1200?

L220--223: Shouldn't the number of epidemic forecasts correspond to 625 times the number of parameter sets?

L229: "but narrow because we ignored uncertainty in the parameters listed in Table 1, and thus should be interpreted with caution" It is also narrow because uncertainty in estimated parameters for each fit is not taken into account (though, in practice, these uncertainties might be all washed away by simulating across multiple parameter sets and may not matter too much).

L230: This should be table 2.

L234--239: Please divide this sentence up---it is difficult to understand. It is also not at all clear what the scoring is trying to measure. For example, why do you only consider values that are exactly equal to the observed values? Why does F suddenly appear? Why does the first summation not have any indices?

L241: It's not clear which quantity is being compared with cases at this point in the text. Please clarify.

L261: "Counterfactuals" This section needs to be explained more clearly and requires a better preface. It is confusing what the purpose of this section is when the authors suddenly talk about diverging counterfactual scenarios (even for a reviewer who has already read through the paper once). It's also not exactly clear what the third counterfactual is trying to achieve (presumably to

show that intervention introduced later will require additional measures to match the effectiveness of the intervention introduced earlier?).

L294--308: I wonder if it's possible for the authors to add RE values in Figure 4 (maybe in the top left corner of each panel) instead of referring to Figure 3. It's difficult to go back and forth two figures to try to figure out the RE values that correspond to the scenarios presented in Figure 4, especially given that Figure 4 is presented at a 3 week interval, whereas Figure 3 is presented at a 1 week interval.

L306--308: I don't see any underestimation...Where is this result presented?

L309: In reality, there are presumably considerable lags between when people develop symptoms and when cases are reported (and there were probably under-reporting of symptomatic cases as well). So I'm surprised that the number of symptomatic infections fits well with reported cases. Any comments from the authors? This is very briefly mentioned in L395 ("lagged one week for reporting lag") but I think it should be explained earlier (in Model assessment section).

L320: "We originally designed and fit our model" => "designed and fitted"?

L407: "This pattern occurred throughout the U.S. during the summer resurgence" any citations?

Table 1: "More states increases the number of parameters and will be harder to fit" increase => increase?

Table 1: "Exponentially distributed rates vs Erlang distributed rates" This sounds like rates are distributed. Please clarify.

Figure 1: "SafeGraph" is never mentioned in the text. Maybe remove unnecessary details to avoid confusion?

Figure S6: axis labels overlap

Referee: 1

Comments to the Author(s).

The authors presented a simple mathematical model of COVID-19, implemented as a stochastic discrete-time model and fit to data using maximization via iterated filtering using the R package pomp. They then evaluated the performance and accuracy of the predictions of the model.

The paper is well-written and the revised version is a big improvement over the authors' initial submission. Many of the original issues with their original manuscript (especially with regard to description of the models and methods) have been resolved. However, I still have some concerns about the convergence of the parameter fitting using mif2. Here are some more specific comments:

1. The most important issue for me is still the lack of parameter profiles. In Figure S6 the authors present the parameter sets versus the likelihood. These could be considered "poor man's profiles." Unfortunately since these are not properly computed profiles, we cannot really use them to find confidence intervals of the parameter. More worryingly, these plots seem to indicate that most parameters do not yet have a properly computed maximum likelihood estimate. Actually running proper profiles may help the authors actually maximize their parameter sets. Profiling is often required to get good maximization which one often cannot get from just running a large number of MIF iterations (since MIF iterations can often get stuck or have small variance after the first few MIF runs, especially with the small number of particles that the authors used, which is often not enough when fitting real data). Big improvements in likelihood can be possible after running profiles across all fitted parameters, so I strongly suggest the authors at least try to run proper profiles and present them in their manuscript. I should note that unfortunately sometimes

proper profiling ends up giving us flat profiles, even when the initial MIF runs yield some curvature. This is important to learn, even if the results end up just showing us that the parameters are very much unidentifiable (usually because the model really does not fit).

2. After doing the profiles, the authors can then use the proper 95% confidence intervals from the profiles instead of using "two log likelihood units" which is a bit arbitrary.

3. If the authors are able to get sharper MLEs for their parameters, the range of susceptible values shown in Figures S8 may change (although unfortunately sometimes these values end up being less realistic). They may also be able to determine which of the two very distinct trajectories in the first panel is a higher likelihood scenario. Otherwise this gives a huge range of estimates on the total number of people that have been infected at any time point. While the authors point out that "majority of parameter sets within for all fits suggested a large percent of the population remained susceptible on the fit date," this is not enough. Because sometimes a majority of MIF runs might end up in the same parameter region, not because that is actually the highest likelihood region but just because of how the gradient of the likelihood surface is near that local maximum. So I would not use the density of the parameter values at the end of the runs as indication of what is "most likely" (also an issue for other figures).

4. I don't think Google Drive is the best place to make results available. Couldn't these have been also uploaded to GitHub? Or use DataDryad?

Author's Response to Decision Letter for (RSPB-2021-0811.R0)

See Appendix D.

Decision letter (RSPB-2021-0811.R1)

26-Jul-2021

Dear Dr Mordecai

I am pleased to inform you that your manuscript entitled "The impact of long-term non-pharmaceutical interventions on COVID-19 epidemic dynamics and control: the value and limitations of early models" has been accepted for publication in Proceedings B.

Data Accessibility section

Open Access

Paper charges

Sincerely,

Professor Hans Heesterbeek

Associate Editor:

Board Member

Comments to Author:

(There are no comments.)

Appendix A

Reviewer’s report on: “The impact of long-term non-pharmaceutical interventions on COVID-19 epidemic dynamics and control: the value and limitations of early models”

In this paper, the authors assess the ability of early SARS-CoV-2 transmission models to accurately estimate the impact of control measures and predict the short-term dynamics of COVID-19 cases and deaths. Specifically, the authors develop a homogeneous SEIR-like model of SARS-CoV-2 transmission and fit it to daily COVID-19 mortality data in Santa Clara county, California using modern statistical inference methods based on iterated filtering [1, 2]. The estimation procedure is repeated to sequentially assimilate more and more data points as the epidemic unfolds, so as to determine if and how the parameter estimates and the model forecasts improve over time. It is found that the model provides accurate short-term (1–2 months) forecasts of COVID-19 deaths, but fails to predict the second wave of COVID-19 cases that started in late June 2020.

This is an interesting, well-written study that uses modern statistical tools to measure the forecasting ability of simple transmission models during an unfolding epidemic. I especially liked the idea of sequentially repeating the estimation to determine how the information accumulates and allows parameter estimation and disease (or death) forecasting. Nevertheless, I have a number of rather substantive comments, which I hope will help improve the manuscript. To summarize, my main concerns are 1) that the impact of non-pharmaceutical control measures is not appropriately modeled, which makes the interpretation of the model’s forecasting ability difficult; and 2) that the authors currently provide no evidence that the estimation procedure has converged to the maximum likelihood estimate, such that it is unclear if the parameter estimates are reliable or not. In addition, I think some parts of the manuscript need to be clarified.

Dr. Matthieu Domenech de Cellès

Major comments

1. [*Modeling of control measures*] In Santa Clara county, as in other parts of the US, a number of non-pharmaceutical control measures were implemented to reduce the spread of SARS-CoV-2. First of all, the information about these measures is scattered throughout the manuscript, so that it is difficult to understand which interventions were implemented and when. On page 12, the authors state that “[...] work-from-home, social distancing, and shelter-in-place orders occurred early in the epidemic.”. I understand that the shelter-in-place order started on March 17, 2020, but what about the other measures? Later in the manuscript (p 15), the authors indicate that “Epidemic dynamics fundamentally shifted in June when Santa Clara County began to relax social distancing orders; [...]” But when did that happen, exactly? Providing accurate information about the timing of these events (for example in a table) would help the reader understand the sequence of control events and how they may have affected transmission dynamics. In any event, the authors model this sequence of control measures using a simple step function for the transmission rate, which is reduced by a factor $1 - \sigma$ (estimated from the data) after March 17. This modeling is (obviously) wrong after the relaxation of control measures in June 2020, which explains why the model fails to capture the second wave of cases during summer. Even before June, it seems the model also imperfectly captures the impact of control measures. Indeed, looking at figure 1A, the variance in the estimate of R_0 does not decrease over time, even though the number of data points incorporated in the model increases. This suggests that the model is misspecified in some way. Hence, I would suggest to consider other ways to model the impact of control measures and to see how that affects forecasting. Indeed, the fact that the model is unable to capture the second wave may not reflect a defect of the model *per se*, but rather a problem in modeling the time-varying transmission rate.
2. [*Convergence of MIF2*] The authors indicate that, for each of the 200 fixed parameter sets, they ran 6 MIF2 estimations to estimate the 3 model parameters (β_0 , σ , and E_0). First, the authors should provide more information about how the starting parameter values were generated (using Latin hypercube sampling, I guess?) and about the algorithmic parameters of MIF2 (cooling type and schedule, random walk standard deviation for perturbations of estimated parameters, etc.). More fundamentally, second, the authors should provide evidence that the algorithm has actually converged to the maximum likelihood estimate, for example by inspecting the slice or the profile likelihood. In my experience, running only 6 replicate estimations of MIF2 may not be enough to reach the MLE. Previous modeling work based on this method typically proceeded in several steps, by first using trajectory matching to get rough estimates and then 50–100 replicate MIF2 estimations [3, 4, 5].
3. [*Calculation of confidence intervals*] Looking at figure 1, I wonder how the confidence intervals were calculated? I understand the variability in parameter estimates reflects the 200 different fixed parameters and, for

each of these, the 6 estimations of MIF2. But, as indicated in the comment above, with only 6 MIF2 replicates, it is unclear if the parametric uncertainty for each fixed parameter set is appropriately captured.

4. *[Quantitative comparison with case data]* The authors argue that, early in the epidemic, case data may be biased and therefore proceed to estimate the parameters using only death data. Previous modeling studies also made this choice [6], which seems entirely reasonable. But why, then, perform a quantitative comparison between the model-based and the observed case data (Figure 3), in addition based on the arbitrary assumption that 10% of symptomatic infections are reported (p 9)? Given the time-varying biases in case data, it seems highly unlikely that the model would capture them very well. I think a qualitative comparison would be sufficient, for example to see if any version of the model is able to capture the second wave.
5. *[Poisson observation model]* The authors use a Poisson model to capture the variability in death reporting. However, figure 1 suggests that this model may not be flexible enough, as the estimates of R_{eff} seem highly sensitive to short-term variations (in particular peaks) in the death data. Why not use a more flexible negative binomial model, as is standard in the field [3, 6]? The extra over-dispersion parameter can be typically very well estimated and will likely improve the estimation of other parameters.
6. *[Distribution of model-based durations]* It would be useful to derive the model-based distribution (not just the mean) of the generation time and of the onset-to-death time. Empirical estimates for these quantities are available [7, 8, 6], and any discrepancy in the variance may allow refining the model structure (for example, using the linear chain trick, by adding more compartments to decrease the variance of the distribution).
7. *[Need for such a detailed model]* Since the goal of the study is to assess the limitations and the accuracy of early models, it seems weird to use such a detailed model. Indeed, much of the information needed to parametrize the model (like the characteristics and the extent of asymptomatic infections) is typically not available early in an epidemic. In this respect, using a simpler SEIR model would make more sense and would also considerably simplify the problem. Admittedly, ignoring asymptomatic infections can cause problems if their duration differs from that of symptomatic infections [9]. Current evidence, however, indicates that the two durations do not differ much [10], and the authors also assume a similar duration (7 days) anyway.

Minor comments

- The description of the model could be improved to help the reader understand its structure (it took me a long time to understand it). First, I suggest to move the model schematic as a figure in the main text. Second, I suggest to present the deterministic variant first (with the corresponding system of ODEs), then the stochastic formulation using the Euler simulation scheme. Finally, some changes in the model's symbols used may be useful. For example, in infectious disease modeling λ is typically used to represent forces of infection (not infectious periods) and this choice may be confusing to some readers.
- Equation 11, p 5: parameter β is missing.
- Table 2: looking at equation 16, it seems that $1 - \delta$ (not δ) represents the fatality ratio among hospitalizations.
- "COVID-19, caused by the emerging virus SARS-CoV-2, has rapidly expanded across the globe, overwhelmed 2 some healthcare systems, and led to nearly a million recorded deaths with the pandemic still underway as 3 of early September, 2020.": need a reference to support that statement.

References

- [1] King AA, Nguyen D, Ionides EL. Statistical Inference for Partially Observed Markov Processes via the R Package pomp. *Journal of Statistical Software*. 2016;69(1):1–43.
- [2] Ionides EL, Nguyen D, Atchadé Y, Stoev S, King AA. Inference for dynamic and latent variable models via iterated, perturbed Bayes maps. *Proc Natl Acad Sci U S A*. 2015 Jan;112(3):719–24.
- [3] King AA, Domenech de Cellès M, Magpantay FMG, Rohani P. Avoidable errors in the modelling of outbreaks of emerging pathogens, with special reference to Ebola. *Proc Biol Sci*. 2015 May;282(1806):20150347.

- [4] Domenech de Cellès M, Magpantay FMG, King AA, Rohani P. The impact of past vaccination coverage and immunity on pertussis resurgence. *Sci Transl Med.* 2018 03;10(434).
- [5] Magpantay FMG, Domenech DE Cellès M, Rohani P, King AA. Pertussis immunity and epidemiology: mode and duration of vaccine-induced immunity. *Parasitology.* 2016 06;143(7):835–849.
- [6] Flaxman S, Mishra S, Gandy A, Unwin HJT, Mellan TA, Coupland H, et al. Estimating the effects of non-pharmaceutical interventions on COVID-19 in Europe. *Nature.* 2020 Jun;.
- [7] Bi Q, Wu Y, Mei S, Ye C, Zou X, Zhang Z, et al. Epidemiology and transmission of COVID-19 in 391 cases and 1286 of their close contacts in Shenzhen, China: a retrospective cohort study. *Lancet Infect Dis.* 2020 Apr;.
- [8] Verity R, Okell LC, Dorigatti I, Winskill P, Whittaker C, Imai N, et al. Estimates of the severity of coronavirus disease 2019: a model-based analysis. *Lancet Infect Dis.* 2020 Mar;.
- [9] Park SW, Cornforth DM, Dushoff J, Weitz JS. The time scale of asymptomatic transmission affects estimates of epidemic potential in the COVID-19 outbreak. *Epidemics.* 2020 06;31:100392.
- [10] Li R, Pei S, Chen B, Song Y, Zhang T, Yang W, et al. Substantial undocumented infection facilitates the rapid dissemination of novel coronavirus (SARS-CoV2). *Science.* 2020 Mar;.

Appendix B

RESPONSE TO EDITOR

First, two referees raised concerns about the lack of information on whether the numerical procedure has converged when fitting the model. This is a very important issue and I think it needs to be thoroughly addressed.

- First, we have expanded our description of how we fit our model, including all of the parameters of the pomp fitting procedure (lines 186–196).
- Second, we have created a series of supplemental plots of model diagnostics including parameter estimates of the three fitted parameters with uncertainty (Figure S2), traces of the likelihood and parameters over iterations of the mif2 fitting algorithm (Figures S4, S5), and scatter plots of the likelihood of the fitted model over the range of values for all sampled parameters (Figure S6).
- Third, we now explain in the text the ways in which model convergence struggled (e.g., lines 282–293) and have more explicitly mentioned issues with parameter identifiability (e.g., line 285, 292).

Second, R2 wondered if the model might be mis specified in some way since certain aspects of the fit do not seem to improve as more data is included.

- First, in the Introduction we now detail the exact purpose for our chosen model structure and highlight that rapidly changing epidemic circumstances can quickly render models obsolete (i.e., mis-specified at some point in time despite originally being appropriate), which we have now made more of a central message of the paper (e.g., lines 70–81, lines 87-90).
- Second, in the Discussion we describe that model fits did not necessarily improve as more data were included because some of our early assumptions became less realistic as time passed and the situation evolved (e.g., lines 414–417).

Third, there is also a general request that more information be given about the details of the model fitting and implementation, to the point where it would be possible for a reader to reconstruct the results if desired.

- First, we have expanded our description of the transmission rate parameter (lines 141–153) as there was some confusion about the time-dependence of transmission rate.
- Second, we have also provided more detail about the mif2 fitting procedure (lines 186–196).
- Third, we have created a new repository with more streamlined code and saved fitted .rds files, as well as saved output from simulations in a publicly available Google Drive folder (files are too large for github), so that any user can quickly recreate our results.

Finally, I agree with R3 that it would be good to include a more complete discussion of other models that address the same issue. I realize that the authors cannot possibly include any formal analysis of other models in a paper of this length but if the goal is to examine how well mathematical models of epidemics perform in the early stages of an outbreak, then I think other models should at least be discussed in as much detail as possible.

- We now more extensively discuss other models and cite many new additional references in the revised Introduction and Discussion, specifically considering the strengths and weaknesses of other early models. Some of this text is dedicated to early epidemic models generally (e.g., Introduction lines 15–24, new Table 1) while other text focuses on COVID-19 models (e.g., Introduction lines 46–67, Discussion: *Early modeling and lessons for future epidemic control*).

RESPONSE TO REVIEWER 1

The paper is well-written and interesting. The goal of the paper is very good. However, I have some concerns about the convergence of the parameter fitting using mif2. This convergence issue is always an important one when using stochastic fitting algorithms and it was not clearly shown by the authors that their parameters had converged. Here are some more specific comments:

- Thank you for your kind words and thorough review.

1. Equations (1)-(10) are written in differential equation form where the transition terms $d_{\{X,Y\}}$ are transition rates. But the terms $d_{\{X,Y\}}$ as defined in equations (11)-(18) are defined for discrete time models. I understand that the continuous time stochastic process was simulated using Euler approximation. However the equations are inconsistent. There are not equations for the continuous time stochastic process. Equations (1)-(10) are written as if they are deterministic differential equations and Equations (11)-(18) are equations for discrete-time stochastic process. The authors need to choose a more consistent way to present their model.

AND

2. Given the inconsistencies of Equation (1)-(10) and (11)-(18), I think it would be better to just put the diagram in the supplementary file in the main text and remove the differential equation system in (1)-(10) which the authors say they did not use.

- Thank you for identifying this inconsistency; we agree that this notation was confusing. As suggested, we moved the model diagram into the main text (now Figure 2) and removed the differential equations (previously 1–10) in order to maintain consistent presentation of the discrete-time stochastic process used in the model. To further streamline the description of our model, we have edited the model diagram so that at each state transition the rates governing movement from one compartment to another are also displayed. In doing so, we have moved Equations 11–18 into the supplement, so that now in the main text the model diagram stands alone as the description of the model, a strategy employed by other papers using discrete-time stochastic processes in pomp (e.g., Magpantay et al. 2016: Pertussis immunity and epidemiology: mode and duration of vaccine-induced immunity; *Parasitology*).

3. The authors explain that they will use a transmission "parameter" β that will change over time. However this β does not appear in equations (1)-(10) or (11)-(18). I assume it is supposed to be part of the expression in equation (11) and was just accidentally dropped from there? This would be the normal location for it and what their online code suggests.

- Thank you for catching this error (which was caused by a problem with a LaTeX macro). We have corrected it to properly include the transmission parameter β .

4. Regarding the parameter values chosen in Table 1. It is important to note that parameters estimated from other sources of data do not often translate to the same way in a population-level model. While I am not suggesting that the authors need to change their parameters, they should note that this can be an important issue. For example when using $\lambda_A=1/5$ days that the authors chose for the asymptomatic rate (I should note that the authors actually wrote in Table 1 that $\lambda_A=5$ days which is inconsistent with equation (13) which clearly shows that λ_A should be a rate and not a period). Since the authors say in lines 74-75 that "We use an Euler approximation of the continuous time process," the binomial draws involving λ_A in equation (13) is supposed to approximate a continuous time process which means λ_A is supposed to be an exponential rate. However, I expect that most observations of the asymptomatic period does not have exponential distribution, but a more "central" distribution, such as an Erlang distribution.

- Thank you for raising these issues. First, we have converted all transition times to rates (e.g., Tables 1 and 2 now use $\lambda_A=1/5$ and the model diagram has been updated accordingly). To be as straightforward as possible, we now explicitly state (lines 178–180) that we use the inverse of the average duration in days as the exponential rate that individuals leave states. Second, we now include the possibility of using Erlang distributed rates in Table 1, explain that we use exponential rates for simplicity in this early model but updated later iterations of the model to use Erlang distributed rates (lines 132–138), and emphasize in the Introduction that our intent in this analysis was to examine an existing, potentially flawed early model

rather than improving upon that early model (lines 82–90). Third, we now include in the Methods the caveat that parameters estimated from other sources may not necessarily be transferrable to population-level models (lines 171–173).

5. In Tables 1-2, the κ parameters are defined to be "relative infectiousness." It is not clear based on these tables or other information provided by the authors what this infectiousness is "relative" to (especially since $\kappa_P, \kappa_M, \kappa_S$ are set to be one)? I guess what they mean is all those κ parameters are equal to 1, while κ_A is the relative infectiousness of the asymptomatic infections (which is incorrectly written as κ_S in Table 2) is the relative infectiousness parameter for asymptotically infected individuals. This should be clarified.

- We have now clarified our language by specifying that the κ parameters are defined relative to presymptomatic infection, which has a κ value of 1 (lines 179–181). We further note that in the absence of data and empirical estimates, we assume κ values of 1 for all but κ_A , which was estimated to be less than one. (As a note we also had κ_A printed incorrectly in Table 2 (now Table 3), which we have now corrected.)

6. Tables 1-2: The λ terms in these tables are written in "days" but from the equations in (11)-(18), it looks like they should be rates. I assume the authors implemented this correctly but

- We have converted all transition times to rates. Tables 2 and 3 now use $\lambda_A=1/5$ and the model diagram has been updated accordingly.

7. In Page 10 the authors provided some details about the settings used in mif2 but there are more parameters involved in the mif2 runs and I needed to check their code to see these other parameters. The authors should mention, perhaps in the supplementary file, if the parameters that they chose resulted in good convergence diagnostics in mif2 (see for example my next comment.)

- Thank you for carefully checking our code and mif2 settings. First, we have greatly expanded the details we provide about our exact mif2 fitting procedure in the Methods section (lines 186–196). Second, we now include more extensive convergence diagnostics in the supplement (see response to following comment).

- Finally, we have created a new repository (that contains just the code relevant to this manuscript) with more streamlined code and saved fitted .rds files as well as saved simulation output in a publicly available Google Drive so that any user can quickly recreate our results (URLs given in the "Data and Code Availability" statement above the references).

8. Lines 97-99: The authors said "we fit parameters to daily deaths for each of the 200 parameter sets using six particle filtering runs with variation in starting values; each run used 100 iterations and 3000 particles." Do they mean "six iterated filtering runs"? And what does "variation in starting values" mean? Does that mean for each parameter set they generated six different initial conditions. Based on their code, it seems like they only allowed for the susceptible class and exposed class to be nonzero at the beginning of their simulation (while the authors mentioned that they fitted the initial exposed class, it was not actually clearly stated in the main text that all others except for the susceptible class were assumed to be zero). Additionally, it is important to explain, perhaps in a supplementary file, if the initial values that mif2 ended up with reasonable values? For example, mif2 might end up with a parameter set suggesting 50% of the population was susceptible and 50% were exposed at the beginning of the simulations, which would not be reasonable. These situations can happen quite often with uncontrolled optimization.

- We have now expanded this section to clarify the exact mif2 fitting strategy that we used. For each parameter set from the Sobol sequence, we fit the model six times using mif2 (six iterated filtering runs), perturbing the starting values for the three parameters we fit ($E_0, \beta_0, \sigma_{SIP}$) using random draws from log-normal or uniform distributions (variation in starting values) (details now given in lines 179-188). Each fit was performed with 300 mif2 iterations ($N_{mif} = 300$) and 1000 particles ($N_p = 1000$), and included a random walk over only E_0, β_0 , and σ_{SIP} , all of which were constrained to be positive (E_0, β_0) or between zero and one (σ_{SIP}).

- As you point out, only the S and E class was assumed to be non-zero at the start of the simulation; we now state this in lines 160-162.

- Finally, we now include a series of supplemental plots as convergence diagnostics: (i) estimated values of the three parameters fit by mif2 (Figure S2), (ii) traces of log-likelihoods during the mif2 procedure for a subset of the parameter sets to show convergence of the log-likelihoods (Figure S4), (iii) traces of the three fit parameters over the mif2 iterations for a sample of parameter sets illustrating both the reasonable final values of parameter sets and the confluence of parameter estimates for different fitting dates for parameters other than E_0 (Figure S5), (iv) plots of parameters and log-likelihood to show whether there was any curvature in the likelihood surface over parameters (Figure S6); and (v) trajectories of percent of population remaining susceptible over time for different fit dates demonstrating the majority of filtering trajectories resulted in reasonable estimates for the last date of the fit and that less plausible estimates were rejected with additional data (Figure S8).

9. The authors' code do not appear to include any constraints for the parameters in Table 2 other than to require them to remain positive during optimization. Did the final values of the parameters resulting from mif2 runs actually remain within the range provided here? If the parameters ended up with values outside of what the authors had in Table 2, this should be mentioned.

- As mentioned previously, during the mif2 optimization, only E_0 , β_0 , and σ_{SIP} , were fit and these parameters were constrained to be positive (E_0 , β_0) or between zero and one (σ_{SIP}), a point we now clarify in the Methods (lines 193–194). Parameters listed in Table 2 (now Table 3) were included in the Sobol sequence and so, by construction, remained within the ranges provided.

- Additionally, in a newly included supplemental plot (Figure S6) we highlight that the best fitting parameter sets contain parameters that are not at the edges of the ranges assumed in Table 2.

- Further, we show (Figure S2) that the estimates for the three fitted parameters follow a reasonably tight distribution to illustrate the general support for a fairly small range of conceivable parameter values.

10. There were no details about how likelihood was computed at the end and I did not see it in the authors' code either (although I perhaps just missed it). Since mif2 does not provide the correct likelihood (unless all random walks were set to zero at the end), I assume pfilter was used to compute the actual likelihood. But I did not see details on how many particles were used for pfilter. I assume the authors would have estimated the likelihood using pfilter (by running pfilter multiple times then taking the logmeanexp) but this was not mentioned anywhere. Also, the authors should mention the spread of the multiple log likelihood estimates given by pfilter to see if the likelihood estimates are reasonable. They should make sure to attach this in their results if they publish the parameter sets

- Thank you for pointing out this unintentional omission. Indeed, pfilter was used after the mif2 function was run to calculate log likelihood. We used 10 replicates of pfilter and logmeanexp to get mean and standard error on the log likelihood estimates. We now state this on lines 188–190. Additionally, we have added a supplemental figure to show the mean and standard errors among log-likelihood estimates for each parameter set (Figure S3).

- Finally, we have adjusted our code to save means and standard errors for the likelihood estimates from the 10 pfilters and have included these in the saved .rds files.

11. No values of the final parameters were presented in the results and no confidence intervals given. No profiles of the parameter values to confirm that the likelihood has actually been optimized. It is unclear if each of the short time intervals that the models were being fitted to provide enough information to get clearly defined maximum likelihood estimates. The authors should at least provide profiles and confidence intervals resulting from fitting to the first and longest time interval.

- We agree that the short time intervals over which the models are fit leads to problems with parameter identifiability. We now reference concerns about parameter identifiability (e.g., line 168, 285, 292) in the main text. Due to these concerns, rather than present values of individual fit parameters (i.e., β_0 and σ_{SIP}), we show the resulting reproduction numbers that are derived from the fit parameters (Figure 3). In addition, we have added supplemental coefficient plots to show the range of values for fitted parameters (Figure S2) and parameter scatter plots versus log-likelihood (for the first, middle and last fitting dates) to illustrate both which parameters the model has the power to best identify and that the log-likelihood is not

showing indications of peaking outside of the specified range for any of the Sobol sequenced parameters (Figure S6).

12. Line 109: The authors mentioned that R_0 was "estimated" to be β_0 divided by duration of the average infection (as defined by their model structure). But it should be straightforward to just compute the exact R_0 , which is the spectral radius of the next generation matrix generated from the differential equation system that is the basis of the continuous time processes. Then this would involve the various relative infectiousness parameters, as well as the presymptomatic duration. This can affect R_0 significantly.

- We believe we have created some confusion with our use of the term "estimated" instead of "calculated" (we have updated the language in this section to use the term "calculated"). While we did not use the spectral radius of the next generation matrix, the transmission rate / infectious period calculation did include the various relative infectiousness parameters.

13. Line 135: How are 50 parameter sets "randomly selected" from the 200 model fits? Are these selections weighted by likelihood? If some parameter sets ended up with higher likelihoods than others, as one might expect from a stochastic fitting algorithm, then parameter sets with higher likelihoods should be weighted more.

- We originally had a threshold of 2 log-likelihood units, above which all fits were used in the CI calculation. However, in most cases many more than 50 fits were above this threshold, which caused some issues with the current way our code was written (RAM problems on local machines). We have rewritten part of this code and sent jobs to a cluster and now include all fits within 2 log-likelihood units of the best fit.

14. In Lines 144-146, how were the confidence intervals for R_0 and R_E computed given that there are no confidence intervals for the parameters that are used to compute them? Were these just taken from the distribution of values at the final mif2 iteration of each of the 1200 runs? Also, since R_E was computed using the "median proportion of the population remaining susceptible," it should be easy to plot the median number of susceptibles generated by the models also and see if these are reasonably consistent with estimates coming from antibody testing, etc.

- We calculate R_E and R_0 separately for each parameter set using the values from the final mif2 iteration for each of the 1200 runs, and present the central 95% interval among those calculated values. As a result, these intervals include uncertainty from the 200 Sobol sequences as well as uncertainty in the fitted parameters for a given Sobol sequence (among the six independent mif2 replicates with different starting values).

- First, we now clarify on lines 207-211 that the uncertainty we draw on parameter estimates include variation in fitted values among replicate mif2 iterations and variation in estimated parameter values across the 200 parameter sets. We further describe the uncertainty bands on forecasts of cases and deaths on lines 223-231)

- Second, we now provide a supplemental figure that plots the change in susceptibles over time to show that our fits produce realistic dynamics (Figure S8).

15. The authors said "We assessed the model forecast performance by computing the mean absolute error of cases and deaths across two weeks after the final date used in the corresponding model fit." This is fine but the authors used Poisson errors when fitting the model, suggesting that they preferred to use a distribution with a variance that scales with the mean. So wouldn't a fairer comparison of the forecast performance would be using Poisson errors instead of absolute errors? Otherwise, why not just use normal errors (equivalent to least squares) from the beginning?

- This is a great point, we have changed our method of quantifying model fit from absolute error to a quadratic score, one of a few strictly proper scoring rules for a predictive model with a Poisson error distribution. We discuss this scoring rule on lines 233-239.

16. The authors should plot the mean number of susceptibles in the population over time to see if the "hidden states" of their model are looking reasonable. In many SEIR models for COVID, the number

of susceptibles drop by large, significant fractions, inconsistent with what antibody testing results suggest.

- We now include supplemental Figure S8 that plots S over time (see our response to Comment 14 for details). While early model fits (from April 1, 2020) found that high depletion of susceptible in the population was feasible among extreme parameter sets, subsequent fits found such rates of susceptible depletion implausible and the majority of parameter sets within for all fits suggested a large percent of the population remained susceptible on the fit date (violin density plot shown in Figure S8).

17. I agree with what the authors say in line 29, "most early models quickly became out-of-date as more sophisticated modeling options became available." and line 39 "it is important to understand the value and limitations of early models and to assess their accuracy over time." I suggest that the authors spend more computational effort on fitting these models by generating the profiles and confidence intervals for each parameter that they are fitting and including the MLE point estimates and confidence intervals in their results. It is important to establish confidence first that the model fitting has converged and the fits are reasonable before using this for forecasting and analysis of the effects of different interventions.

- We now include a series of supplemental plots of parameter estimates (coefficient and pairs plots) and model diagnostics (likelihood traces) to show model convergence (Figures S2-S6).

1. Lines 60-61: The citations are not in increasing order.

- Thank you for noticing this issue, and we have corrected it by renumbering them in the order in which they appear.

2. Equations (11)-(18): The authors should mention what dt is before using it. I understand from the text that dt = 4 hours, but the symbol should be defined first anyway.

- Given that we have now moved all equations to the supplement, we define *dt* directly after the printing of those equations.

3. Equations (11)-(18): Some of the transition terms here are written as multinomial draws and some as binomial draws. In the multinomial draws, the "transitions" into the same compartment (basically the number that stay in its compartment, such as $d_{E,E}$ for example) is written out explicitly even though this "transition" is not listed in equations (1)-(10). Such transitions are not written out explicitly for the binomial draws. I suggest the authors make a decision on how to treat these "transitions" and use a consistent notation in their equations.

- We have simply dropped the non-transition terms (as Reviewer 3 also requested).

4. Line 126: What does "from the single best fit on April 22 across the 200 parameter sets as defined by negative log likelihood" mean? Do the authors just mean they the parameter set with the best likelihood? I am also still wondering about where how the likelihoods were computed (using pfilter?) and how the best one was found. Weren't they six runs of each of the 200 starting parameter sets (each run having different initial conditions?) so do they mean the parameter set with the best resulting likelihood estimate across the 1200 runs?

- Yes, we did mean the best parameter set with the best likelihood; we have clarified this in the text. Also, as we mentioned earlier, we did use pfilter, but now explicitly state this in the Methods.

5. One important non-pharmaceutical intervention that the authors did not explicitly include in the model is contact tracing and quarantine. This should be mentioned.

- We now briefly mention that we do not consider contact tracing and quarantine on lines 252-253 of the Methods.

RESPONSE TO REVIEWER 2

1. [Modeling of control measures] In Santa Clara county, as in other parts of the US, a number of non-pharmaceutical control measures were implemented to reduce the spread of SARS-CoV-2. First of all, the information about these measures is scattered throughout the manuscript, so that it is difficult to understand which interventions were implemented and when. On page 12, the authors state that “[. . .] work-from-home, social distancing, and shelter-in-place orders occurred early in the epidemic.” I understand that the shelter-in-place order started on March 17, 2020, but what about the other measures?

- Thank you for pointing out this confusion. To streamline the dates and context for the model in Santa Clara County, we have created a "Timeline of Events" figure (now Figure 1) that shows epidemiologically important dates (public health initiatives/orders/outcomes) alongside model development and writing. We have also revised the sentence you highlight to include the words "in the first half of March."

2. Later in the manuscript (p 15), the authors indicate that “Epidemic dynamics fundamentally shifted in June when Santa Clara County began to relax social distancing orders; [. . .]” But when did that happen, exactly? Providing accurate information about the timing of these events (for example in a table) would help the reader understand the sequence of control events and how they may have affected transmission dynamics.

- We now include this information in Figure 1. We also updated the given sentence to provide specific dates ("Epidemic dynamics fundamentally shifted in June as Santa Clara County relaxed social distancing orders in phases on May 4th and June 5th").

- Given some amount of confusion expressed by multiple Reviewers on what was fit in the model for the strength of the intervention and transmission rate, we have also clarified how we use three distinct transmission rates for three assumed periods of contact structure (pre intervention, work-from-home, shelter-in-place) (lines 145–153).

3. In any event, the authors model this sequence of control measures using a simple step function for the transmission rate, which is reduced by a factor $1 - \sigma$ (estimated from the data) after March 17. This modeling is (obviously) wrong after the relaxation of control measures in June 2020, which explains why the model fails to capture the second wave of cases during summer. Even before June, it seems the model also imperfectly captures the impact of control measures. Indeed, looking at figure 1A, the variance in the estimate of R_0 does not decrease over time, even though the number of data points incorporated in the model increases. This suggests that the model is misspecified in some way. Hence, I would suggest to consider other ways to model the impact of control measures and to see how that affects forecasting. Indeed, the fact that the model is unable to capture the second wave may not reflect a defect of the model per se, but rather a problem in modeling the time-varying transmission rate.

- First, we completely agree that the step function we used is a strong simplification of rapidly changing human contact patterns. This was an early modeling decision made to accommodate estimating key parameters (e.g., transmission rates during initial and first-wave control periods) given very sparse data. We have overhauled the presentation of the Introduction and Discussion sections to make it clearer that the goal of this paper is to assess the value and limitations of early epidemic models by using our own early model as a case study (in particular, lines 82–90). We have expanded the Introduction to include more context about early modeling decisions, the time frame of our model, and the specific goals of our model. We now also explicitly state that the early model we analyze here gave way to a more sophisticated approach using continuous human mobility, but that the purpose of this paper is to serve as a "snapshot in time" (a case study) analysis of an early model. Indeed, the model does in fact become misspecified after the constant scaling factor is no longer appropriate to describe reality, but that is a shortcoming we hope to highlight, which we now do with reference to our more complicated model now published in *Epidemics*. We also explicitly address the point that uncertainty about R_E declined and model accuracy increased with time until ~ late May, then accuracy began declining again as the situation

changed; i.e., the tradeoff between more time to observe dynamics vs. more time for dynamics to change (lines 388–393, lines 401–407, lines 414–426).

- We agree that it is initially somewhat concerning that the variance of R_0 estimates do not decrease with additional data. However, similar to initial value parameters, as the unmitigated epidemic with R_0 was only in effect for a short time period (from the introduction of the virus until early March when work-from-home began), most information contained within the time series that informs our estimates of R_0 comes from the initial few weeks of data. Given our current model structure, additional data as time passes contributes primarily to certainty in estimates on R_E , whose variance declines rapidly after late April.

4. [Convergence of MIF2] The authors indicate that, for each of the 200 fixed parameter sets, they ran 6 MIF2 estimations to estimate the 3 model parameters (β_0 , σ , and E_0). First, the authors should provide more information about how the starting parameter values were generated (using Latin hypercube sampling, I guess?) and about the algorithmic parameters of MIF2 (cooling type and schedule, random walk standard deviation for perturbations of estimated parameters, etc.).

- Thank you for pointing out this omission. We now define the Sobol sampling procedure that we used for the fixed parameter sets (Methods, line 174) and have now greatly expanded the section on our mif2 fitting procedure to include details about all of the parameters we used (lines 186-196).

5. More fundamentally, second, the authors should provide evidence that the algorithm has actually converged to the maximum likelihood estimate, for example by inspecting the slice or the profile likelihood. In my experience, running only 6 replicate estimations of MIF2 may not be enough to reach the MLE. Previous modeling work based on this method typically proceeded in several steps, by first using trajectory matching to get rough estimates and then 50–100 replicate MIF2 estimations [3, 4, 5].

- First, it appears that some of the language regarding our mif2 parameters was confusing. To be more explicit, we have now clarified in the manuscript that for each of 200 parameter sets created with a Sobol sequence we ran six individual complete mif2 runs, each with random starting points for the fit parameters using 300 iterations ($N_{mif} = 300$) and 1000 particles ($N_p = 1000$) (lines 186–196). After reading this comment and the three suggested references, we believe that our procedures are in line with best practices in mif2 usage.

- Second, we have now added supplemental figures to address concerns about algorithm convergence. In particular, we demonstrate that (1) each mif2 converged to its local maximum (Figure S4); (2) the fit parameters show good convergence across the six mif replicates for each set of fixed parameters from the Sobol sequence across different fitting dates (Figure S5); (3) most parameters fit with mif2 show clear curvature in the log-likelihood surface and while many of the fixed parameters from the Sobol sequence show minimal curvature in the log-likelihood surface, none showed clear increasing patterns at the extremes of the sampled ranges (Figure S6).

6. [Calculation of confidence intervals] Looking at figure 1, I wonder how the confidence intervals were calculated? I understand the variability in parameter estimates reflects the 200 different fixed parameters and, for each of these, the 6 estimations of MIF2. But, as indicated in the comment above, with only 6 MIF2 replicates, it is unclear if the parametric uncertainty for each fixed parameter set is appropriately captured.

- We now explicitly state what is included in our uncertainty bands for cases and deaths in the Methods on lines 220-231). Additionally, given the convergence of the fit parameters (aside from the initial value parameter E_0) for each fixed parameter set (see newly added Figure S5), we believed that the six mif2 replicates captured the range of uncertainty in fit parameters for each set of fixed parameters.

7. [Quantitative comparison with case data] The authors argue that, early in the epidemic, case data may be biased and therefore proceed to estimate the parameters using only death data. Previous modeling studies also made this choice [6], which seems entirely reasonable. But why, then, perform a quantitative comparison between the model-based and the observed case data (Figure 3), in addition based on the arbitrary assumption that 10% of symptomatic infections are reported (p 9)? Given the

time-varying biases in case data, it seems highly unlikely that the model would capture them very well. I think a qualitative comparison would be sufficient, for example to see if any version of the model is able to capture the second wave.

- This is an excellent point. We have followed your suggestion and reframed our assessment of model performance to focus on a quantitative comparison to deaths and a visual, qualitative comparison of the trajectory of model predictions to the trajectory in cases.

- We have reorganized what was previously Figures 3 and 4 into two new figures: Figure 4 now shows our projections for deaths along with the model evaluation (quadratic score) for those projections; Figure 5 shows our projections for cases with no quantitative comparison.

8. [Poisson observation model] *The authors use a Poisson model to capture the variability in death reporting. However, figure 1 suggests that this model may not be flexible enough, as the estimates of Re_f seem highly sensitive to short-term variations (in particular peaks) in the death data. Why not use a more flexible negative binomial model, as is standard in the field [3, 6]? The extra over-dispersion parameter can be typically very well estimated and will likely improve the estimation of other parameters.*

- Thank you for this suggestion. While later versions of this model used a negative binomial observation process for exactly the reasons you suggest here (Kain et al. 2021, *Epidemics*), in this manuscript we aimed solely to evaluate the pre-existing early model, which at the time of initial development included a Poisson observation model to limit the number of parameters that needed to be estimated. We now emphasize this aim in the Introduction (lines 82–90).

9. [Distribution of model-based durations] *It would be useful to derive the model-based distribution (not just the mean) of the generation time and of the onset-to-death time. Empirical estimates for these quantities are available [7, 8, 6], and any discrepancy in the variance may allow refining the model structure (for example, using the linear chain trick, by adding more compartments to decrease the variance of the distribution).*

- We now derive and show the hypoexponential distribution of the onset-to-death time in the online supplement (Figure S1) and compare them to the estimates from citations 6, 7, and 8 that you list here (lines 184–185). We also discuss the linear chain trick to get Erlang distributed rates (which we used in our follow up paper: Kain et al. 2021, *Epidemics*), but given our intent to study the pre-existing early model, state that we did not use this approach in this first model (Table 1, lines 134-135).

10. [Need for such a detailed model] *Since the goal of the study is to assess the limitations and the accuracy of early models, it seems weird to use such a detailed model. Indeed, much of the information needed to parametrize the model (like the characteristics and the extent of asymptomatic infections) is typically not available early in an epidemic. In this respect, using a simpler SEIR model would make more sense and would also considerably simplify the problem. Admittedly, ignoring asymptomatic infections can cause problems if their duration differs from that of symptomatic infections [9]. Current evidence, however, indicates that the two durations do not differ much [10], and the authors also assume a similar duration (7 days) anyway.*

- We agree that much of the information needed to parameterize this model was unavailable early in the epidemic and that it is potentially more detailed than necessary to simply match the trajectory of the early epidemic. However, as we aimed to use this model to evaluate different non-pharmaceutical interventions—including symptomatic isolation—these details were necessary. We now explain how this intended model purpose determined the level of detail and compartments required for the model (Introduction, lines 91–112).

The description of the model could be improved to help the reader understand its structure (it took me a long time to understand it). First, I suggest to move the model schematic as a figure in the main text. Second, I suggest to present the deterministic variant first (with the corresponding system of ODEs), then the stochastic formulation using the Euler simulation scheme. Finally, some changes in the

model's symbols used may be useful. For example, in infectious disease modeling λ is typically used to represent forces of infection (not infectious periods) and this choice may be confusing to some readers.

- We have moved the model diagram into the main text and edited it so that each transition contains the rate used in the binomial/multinomial probabilities. Given confusion from other Reviewers about the equations we have opted to drop equations 1-10, move equations 11–18 into the online supplement, and rely on the model diagram and a verbal description of the model for the main text (similar to the strategy in Magpantay et al. 2016: Pertussis immunity and epidemiology: mode and duration of vaccine-induced immunity; *Parasitology*). For consistency with our other published work that is an expansion upon this model with the same primary authors (Kain et al. 2021, *Epidemics*), we have decided to keep the lambda notation for infectious periods. We do, however, now explicitly state in the Methods that unlike other models that use lambda for forces of infection, we use it here to refer to infectious periods (lines 178–179).

Equation 11, p 5: parameter β is missing.

- Thank you for pointing this out, we have now corrected this error (which was caused by a problem with a LaTeX macro).

Table 2: looking at equation 16, it seems that $1-\delta$ (not δ) represents the fatality ratio among hospitalizations.

- This has been corrected, thanks for noticing this error.

“COVID-19, caused by the emerging virus SARS-CoV-2, has rapidly expanded across the globe, overwhelmed 2 some healthcare systems, and led to nearly a million recorded deaths with the pandemic still underway as 3 of early September, 2020.”: need a reference to support that statement.

- Citation to Johns Hopkins dashboard added.

RESPONSE TO REVIEWER 3

1. The paper lacks discussion of other early models that were used during the current pandemic. Here, the authors consider a stochastic SEIR model that they developed... I am not at all suggesting that the authors should re-do the analysis, but I think it is important to discuss other modeling approaches (are they broadly similar? or different? do same principles/conclusions still apply?).

- This is a great point. We have expanded our discussion of other early models and their strengths and weaknesses in the Discussion, and we now include many additional references. Some of this text is dedicated to early epidemic models generally (e.g., Introduction: paragraph 2) while other text focuses on COVID-19 models (e.g., Introduction: paragraph 4, Discussion: paragraph 5).

2. If possible, it would be nice if the authors can comment on possible remedies for early models and tradeoffs. For example, new models are almost always needed as we learn more about the disease, but this can be time-consuming. It can also be difficult to compare results from new models with those from earlier models. On the other hand, one could imagine building a very flexible model earlier on, but such a model may have too little power to tell us anything useful. I believe such discussion may make the argument of the paper stronger and can further highlight the importance/limitations of early models.

- We agree that this is an important focal point of our work, which we have worked to further emphasize in our revisions. We have added Table 1, which outlines different modeling choices, options, considerations and tradeoffs, and our decisions. With our restructured Discussion, these topics are now the primary focus of the section titled "Early modeling and lessons for future epidemic control" (lines 413–447). In this section we aim to both reflect on what we learned from our early model that is of general interest and to give suggestions for early models for future epidemics.

3. I was confused about the order in which the results are presented. The authors present the scenario analysis first and then discuss their parameter estimates and model predictions. I think it should be the other way around. Model fits and their accuracy should be shown first (Figures 3 and 4). Estimates of the effective reproduction number should be shown after the model fits to assess the validity of these estimates. The scenario analysis should be presented at the end after the validity of the model fits has been demonstrated.

- This is a good point. We have altered the order of the Results so that the model fits and their accuracy are shown before the scenario analyses.

Abstract: R_0 is defined as the basic reproduction number but R_E is not defined ("our estimated RE varied"). Maybe better to be consistent and say effective reproduction number (RE)? But not necessary at all...

- We appreciate this suggestion and have added that R_E is the effective reproduction number.

Line 9: It would be nice to provide some citations of early models that did this and perhaps briefly discuss them to provide a better overview of early models here.

- We have added a few citations for this sentence of early models that informed policy.

Line 14 ("Early models can and should inform policy decisions during an emerging pandemic"): While I agree with the authors, they have not explained why they "should" inform policy decisions at this point. The advantages of early models could be highlighted more explicitly.

- In our rewritten Introduction, paragraph two now focuses on the ways in which early models are important and what roles they can play in policy decisions (lines 15-24).

Line 31: I wonder if it's possible to give some sort of references or examples of doing this...

- In our rewritten Introduction we have reframed the idea of models being used past their expiration date into a much broader discussion about early model assumptions and a changing epidemiological environment. Thus, the sentence you reference has been removed.

Line 78: I wonder why Poisson was used instead of the negative binomial?

- As we also mentioned in a comment to Reviewer 2: later versions of this model used a negative binomial observation process (Kain et al. 2021, *Epidemics*). However, in this manuscript we aimed solely to evaluate the pre-existing early model, which at the time of initial development included a Poisson observation model to limit the number of parameters that needed to be estimated. We now emphasize this aim in the Introduction (lines 82–90) and highlight this choice in Table 1.

Equations: I think it could be useful to show the actual model diagram in the main text rather than in the supplementary file given that the equation is a bit dense. For example, it is not clear from the writing that presymptomatic individuals can be either mildly or severely symptomatic. It's also not entirely trivial to go back and forth between equations 1-10, equations 11-18, and tables 1-2...

- Thank you for this suggestion. We have moved the model diagram into the main text and edited it so that each transition contains the binomial or multinomial probabilities to be explicit about the transitions. Given confusion from other Reviewers about the equations we have opted to drop equations 1–10, move equations 11–18 into the online supplement, and rely on the model diagram and a verbal description of the model for the main text (similar to the strategy in Magpantay et al. 2016: Pertussis immunity and epidemiology: mode and duration of vaccine-induced immunity; *Parasitology*).

Equation 11: It's missing a time-varying transmission rate parameter. It only has relative infectiousness right now...

- Thank you for pointing this out. We have corrected our previous mistake (a typo issue with a LaTeX macro) to properly include the transmission parameter β .

Equations 12-16: I believe $dE, E, dI_p, I_p,$ and dI_s, I_s terms are not needed in the equations (because the authors are drawing a multinomial from E to I_a and I_p); these terms also don't appear in equations 1-10. The current formulation suggests that the authors are drawing three quantities from E (going from E to E, E to $I_p,$ and E to I_m). I suspect this is just a notation problem, and not an implementation problem though...

- This is indeed a notation problem and has been corrected.

Table 1-2: Table 2 caption reads "Parameter range estimates that are not location-specific" which could imply that Table 1 estimates are location specific? I can see that some parameters, such as time from symptom onset to hospitalization, are location-specific but others, such as incubation periods, are probably not. Also, table 2 should say κ_A rather than κ_S . I also wonder why point estimates were used for some and not for others. I suspect this is due to the location specificity of some parameters, but I think it needs to be explained more clearly.

- The "location-specific" phrasing was incorrect, thank you for catching it. We have removed this phrasing.

Line 92: I could not understand how the transmission rate is modeled here. It's briefly explained in Lines 109-112, which reads as if different sigma values are estimated for each week. I could not figure this out until I got to Line 225 ("especially when assuming a constant impact of non-pharmaceutical interventions, as this model does"). The duration of shelter-in-place should also be noted at this point.

- First, we have greatly expanded the description of σ in the Methods section (lines 141–153) and explicitly state that we fit a single value for σ , which assumes a single proportional reduction in β as a function of the reduction in contacts relative to baseline. Further, we state that because we assume a stepwise function for contact patterns with three discrete levels (baseline, work-from-home, and shelter-in-place; these are now given dates on the Figure 1 timeline and a text description in lines 141–153), our model assumes only three possible transmission rates by an infected individual. Your comment also made us notice that we somehow lost a parameter from Table 3—the effectiveness of the early "work from home" Bay Area initiative to reduce contact rate before the Shelter in Place began; it has now been added back into Table 3.

Line 119: I suspect the results can vary depending on how much social distancing is relaxed vs how much transmission rates for symptomatic individuals are reduced. This should be explained more clearly.

- Indeed, this is what we attempted to highlight in Figure S2 (what is now Figure S9), but which we failed to discuss clearly in the text. We now reference this figure with additional main text Discussion on lines 337–344.

Line 127: I think it would be informative to provide what summary statistics you're considering at this point.

- We now list up-front what these summary statistics are.

Line 133: It should be noted that uncertainties associated with beta0 and sigma are also not taken into account. I understand that there are uncertainties in beta0 and sigma due to 200 parameter sets that the authors consider but each fit has its own parameter uncertainty. It would be also nice to cite Elder et al. 2006 here (<https://doi.org/10.1073/pnas.0600816103>).

- We now take into consideration the uncertainties associated with β_0 and σ by including the variation in their estimates across six replicate mif2 runs at each parameter set. Notably, the variation between fit parameters (aside from the initial value parameter E_0), was small as shown for a sample of fixed parameter sets (newly added Figure S5). We also now cite Elder et al. 2006 (in the Introduction).

Line 139: It should be noted that this does not take into account the delay between symptom onset and reporting of cases (and so the predicted cases can be systematically biased).

- We have included a mention of this lag in both the results (Figure 5 caption) and Discussion (line 395). Further, per a request from Reviewer 2, we have switched our quantitative model assessment to focus just on deaths, converting our assessment of cases to a qualitative one (primarily because of the arbitrary assumption of the 10% reporting rate).

Figure 1: Are these CIs also based on 200 parameter sets (as opposed to something like profiling)?

- The CI were based (and are now based) on all fits within 2 log-likelihood units of the best fit. We explicitly state how our uncertainty bands on deaths and cases were calculated on lines 220–231 of the Methods. Further, we describe the CI on the parameter estimates on lines 207–211.

Figure 1: It might be useful to show the daily number of infections for the readers to assess estimates of R_e .

- We now include a timeline of cases in Figure 3.

Line 148: Is this based on any random "one parameter set" or the best fit?

- The best fit, we now clarify this.

Line 151: What is meant by the number of concurrent infections? Does it include everyone who has not recovered including the hospitalized individuals?

- Yes; however, given the confusion surrounding this quantity, we have now switched to using new daily symptomatic infections (the closest our model comes to reported cases) both here (now lines 331, 335) and in the associated figure (Figure 6).

Figure 2: It seems like predictions are made by simulating epidemics beginning from the initial time without conditioning on the data. Instead, predictions should be made by conditioning on the data first, and then simulating forward. The first step can be done by using a pfilter function and taking the particle filtered trajectories. This will remove simulations that do not match the data (e.g., some simulations reach <10 cumulative deaths as of June 1st) and reduce uncertainty accordingly.

- Thank you for pointing this out. We have taken your advice and now use independent pfilters to get particle filtered trajectories and forecast from these for all forecasts and trajectories included in the manuscript. Details given on lines 213–220.

Line 154: "removing an additional 80% and 70% of infective contacts from severe and mild infections respectively" infective -> infective?

- Fixed.

Line 183: I'm a bit confused by how REs are compared here. Since a constant reduction in transmission rate is assumed throughout the shelter-in-place period, the model fit based on the data until July estimates that RE for earlier dates (e.g., early April) is also around 0.8. On the other hand, earlier fits estimate that RE in April is much lower. It would be nice for the authors to comment on this discrepancy and how one should go about interpreting such estimates. For example, when the model is fitted to data until July, are REs for earlier weeks re-estimated as well? Or are previous RE estimates accounted for in the new fits somehow?

- First, yes, each fit is independent and includes data up until a given ending date, which means that R_E near the start of the epidemic can change quite a lot with the addition of a week of data. As we mentioned above, we have now clarified how the R_E values are estimated with a stepwise function with three discrete levels of contact rates. Previously R_E estimates have no effect on subsequent model fits, which re-estimate the same set of parameters. Additionally, we discuss the jump in R_E estimates in late April in the Results on lines 276-282 and in the Discussion on lines 379-382. More generally, we discuss the broader topic of model estimates as a function of increasing data and divergence from initial assumptions starting on line 408 of the Discussion.

Line 183: "Following a spike in deaths, RE increased to 0.977 on April 22" Shouldn't RE increase before a spike in deaths because REs reflect infections?

- Good point. As you say, this is how it would operate in reality. What we meant to convey here was the reason for why the model *estimate* of R_E jumped between these dates, namely that the inclusion of another week of data in which the number of deaths was higher led the model estimating a higher R_E . Yes, there is some lag here, but with so little data and because we are only fitting to deaths, the extra week of data with higher deaths has the effect of increasing R_E estimates. We have rewritten this sentence to make this clearer.

Line 184: Again, forecasts should rely on filtered trajectories, rather than based on unconditional simulations.

- Updated to use filtered trajectories.

Figure 4 caption: "Mean absolute error for Colors indicate the percentage of errors that are over-estimates, where red indicates a high percentage of overestimates and blue indicates a high percentage of under-estimates" Typo at the beginning of this sentence ("Mean absolute error for")?

- Good catch. This extra errant clause has been removed.

Line 284-288: I was unsure whether this discussion was about the model presented in the paper or about other early models in general.

- The text and ideas in the section you highlight has been incorporated into our rewritten Discussion. In response to a more general concern about separating our discussion about our model vs. early models generally, we have separated the Discussion into two sections. In the first section we focus on the success and failures of our model, and only broaden out to a general comment in the last sentence of each of these paragraphs. Then, in the second section, titled "Early modeling, the unfolding COVID-19 pandemic, and lessons for future epidemic control" we broaden out into a more general discussion.

I wonder if the code provided is the correct code? It seems like multiple boxes were used for each infection class and mobility data were also included somehow, but I don't see any descriptions in the paper...

- Another good catch. The code used in this paper is contained in that repository, but was a bit buried by code for continued model development. We have created a new repository with code only relevant to this paper and now identify the correct repository in the Code and Data Availability section.

Appendix C

Reviewer’s report on: “The impact of long-term non-pharmaceutical interventions on COVID-19 epidemic dynamics and control: the value and limitations of early models”

I thank the authors for their thorough and careful revisions. All my major comments have been addressed, and I only have a few minor comments left.

Dr. Matthieu Domenech de Cellès

Minor comments

- Table 1: though interesting, I think this table could be moved to the supplement. In addition, some parts of the table should be revised for more accuracy:
 - Analysis framework: the authors indicate Approximate Bayesian Computation (ABC) for estimation in a Bayesian framework. However, the Bayesian equivalent of MIF2 is arguably the particle MCMC algorithm (pMCMC, also implement in pomp)—not ABC, which, unlike mif2 and pMCMC, is a feature-based estimation method (see Table 2 in Ref. [1]).
 - Observation process distribution: “Possible difficulty of fitting negative binomial dispersion parameter”. As I wrote in my first review, in my experience this is typically not true and this parameter can be estimated very well. Admittedly, this parameter is more important to consider in deterministic model variants, where it allows capturing noise in the data.
 - Transition rates between compartments: in all rigor, it is the duration spent in a given state that follows an Exponential or an Erlang distribution (not the rates).
 - Choosing which parameters to fit and which to fix: to the best of my knowledge, I don’t know an occurrence of a model in which all the parameters were fitted from the data.
- Figure 2: as the parameters indicate the rates of transition between compartments, in all rigor β_i should be replaced by the force of infection.
- The authors use the adjective “hypoexponential” to describe the distribution of the onset-to-death time. Although I understand what is meant here, this adjective is not very common and could be replaced by something more explicit (like the actual name of the distribution).
- Figure S6: I think it would be more effective to use a scatterplot matrix to represent the estimates. Also, please provide the definition of relative log-likelihood in the legend. (If that simply represents the difference with the maximum log-likelihood, “scaled log-likelihood” is more commonly used to refer to that quantity.)

References

- [1] King AA, Nguyen D, Ionides EL. Statistical Inference for Partially Observed Markov Processes via the R Package pomp. *Journal of Statistical Software*. 2016;69(1):1–43.

Appendix D

RESPONSE TO REVIEWER 1

1. The most important issue for me is still the lack of parameter profiles. In Figure S6 the authors present the parameter sets versus the likelihood. These could be considered "poor man's profiles." Unfortunately since these are not properly computed profiles, we cannot really use them to find confidence intervals of the parameter. More worryingly, these plots seem to indicate that most parameters do not yet have a properly computed maximum likelihood estimate. Actually running proper profiles may help the authors actually maximize their parameter sets. Profiling is often required to get good maximization which one often cannot get from just running a large number of MIF iterations (since MIF iterations can often get stuck or have small variance after the first few MIF runs, especially with the small number of particles that the authors used, which is often not enough when fitting real data). Big improvements in likelihood can be possible after running profiles across all fitted parameters, so I strongly suggest the authors at least try to run proper profiles and present them in their manuscript. I should note that unfortunately sometimes proper profiling ends up giving us flat profiles, even when the initial MIF runs yield some curvature. This is important to learn, even if the results end up just showing us that the parameters are very much unidentifiable (usually because the model really does not fit).

- We have now computed likelihood profiles for the three fitted parameters and present them in the manuscript (Methods **lines 215-222**; Results **lines 317-323**; the profiles themselves in **Figure S2, bottom panel**). To profile over a given fit parameter, we selected 30 uniformly spaced points (hereafter, fixed points) and 200 Sobol-sequenced parameter combinations. For each fixed point, we fit the model using the same *mif2* settings as for other model fitting using 3 replicate *mif2* runs each with random starting values for the *mif2* algorithm (3 rather than the usual 6 *mif2* replicates were used due to computational cost) and identified the maximum log-likelihood among the resulting 600 *mif2* parameter sets for that fixed point (these details are described on **lines 215-222**). Given the extensive computational cost of profiling in the presence of 7 Sobol-sequence sampled parameters (18,000 total model fits for one parameter's profile), we have profiled the three fit parameters for just the three primary dates used in our in-depth convergence diagnostics presented in the supplement (April 1, May 13, and June 24, 2020).

- The profiles show clear curvature (despite some jaggedness) in β_0 and σ_{SIP} for the two later fit dates (**Figure S2, bottom panel**). For the first fit date (April 1), the model lacked the information to properly distinguish between different scenarios, reflected in the relatively flat likelihood profiles for all three parameters as well as the uncertainty in R_E (**Figure 3**) and wide range in future simulations (**Figure 4**). For all three fit dates, the profile over E_0 is relatively flat, smooth, and has a high log-likelihood over the range of E_0 , which points to the relatively low importance of that parameter as compared to others (we discuss more about E_0 in our response to Reviewer 3's point 3). We discuss the findings from these parameter profiles in the Results (**lines 317-323**).

- For the Sobol-sequence sampled parameters we have isolated the maximum likelihood for each sampled value for each parameter and have re-plotted these curves in the same way as for the profiles for the fitted parameters in **Figure S7**.

2. After doing the profiles, the authors can then use the proper 95% confidence intervals from the profiles instead of using "two log likelihood units" which is a bit arbitrary.

- As discussed above, we have added a panel to **Figure S2** showing the profiles for the three dates (April 1, May 13, and June 24, 2020). We have kept the three upper panels showing the parameter sets within 2 log likelihood units for those and the remainder of the dates to show their relative equivalence (despite a lack of profiles for the other 10 fit dates).

- It is not feasible for us to profile over all of the non-fitted Sobol sequenced sampled parameters. (We sampled over these parameters because we didn't think there was enough information to fit them and as a way to capture uncertainty in them---at the time there wasn't enough knowledge about them for us to come up with any reasonable point estimates.) Given our use of fitted, sampled, and fixed parameters, as well as our presentation of central intervals on quantities and simulations that result from a combination of the uncertainty in these parameters, we have retained our current method of constructing central intervals on trajectories and reproductive numbers. Thus, while somewhat arbitrary, we believe that our

use of 2 log likelihood units to select realistic Sobol sequenced and fitted parameter sets for central intervals for projections/forecasts is a reasonable compromise.

3. If the authors are able to get sharper MLEs for their parameters, the range of susceptible values shown in Figures S8 may change (although unfortunately sometimes these values end up being less realistic). They may also be able to determine which of the two very distinct trajectories in the first panel is a higher likelihood scenario. Otherwise this gives a huge range of estimates on the total number of people that have been infected at any time point. While the authors point out that "majority of parameter sets within for all fits suggested a large percent of the population remained susceptible on the fit date," this is not enough. Because sometimes a majority of MIF runs might end up in the same parameter region, not because that is actually the highest likelihood region but just because of how the gradient of the likelihood surface is near that local maximum. So I would not use the density of the parameter values at the end of the runs as indication of what is "most likely" (also an issue for other figures).

- Indeed, our fits cover a large range of possible scenarios, especially for the April 1 fit. Specifically, the model finds both biologically sensible epidemic dynamics (e.g., an R_E near 3.5) and explosive dynamics depleting most susceptible individuals equally plausible scenarios (leading to the bifurcation in the susceptible depletion figure; **Figure S12**). The profiles help to show this: 1) the jagged peak in scaled log-likelihood with a very high baseline transmission rate (**Figure S2, bottom left panel, blue line**); 2) the cluster of points with the highest likelihood having very high R_0 and very low R_E (**Figure S7, R_0 and R_E panels, blue points**). Though these points correspond to only a small number of parameter sets that result in a scenario of rapid susceptible depletion (bottom lines in **Figure S12**), the log likelihoods are comparable to the more realistic scenario. Given the proximity in likelihood of other parameter sets and the very limited (and potentially under-counted) data observed by April 1st (**Figure 4, top left panel**), we interpret this to be the result of an overly complex model fit with too little data as early as April 1---we have modified the Discussion (**lines 475-483, lines 493-499**) to further reflect on this shortcoming of our early model.

- Further, we agree that the end location of the *mif2* runs may not be representative of the highest likelihood region. Unless noted otherwise, we limit to resulting parameter sets within the top 2 log-likelihood units when presenting results to help address this concern—we believe the use of Sobol sequencing over the most uncertain parameters and the use of 2 log-likelihood units for the purpose of forecasts (if a bit arbitrary) achieves the goal of capturing a reasonable range of uncertainty in the trajectory of the unfolding pandemic.

4. I don't think Google Drive is the best place to make results available. Couldn't these have been also uploaded to GitHub? Or use DataDryad?

- This is a good point. We have uploaded all files to DataDryad and included the URL in the "Data Availability" section of the manuscript.

RESPONSE TO REVIEWER 2

Table 1: though interesting, I think this table could be moved to the supplement. In addition, some parts of the table should be revised for more accuracy:

– *Analysis framework: the authors indicate Approximate Bayesian Computation (ABC) for estimation in a Bayesian framework. However, the Bayesian equivalent of MIF2 is arguably the particle MCMC algorithm (pMCMC, also implement in pomp)—not ABC, which, unlike mif2 and pMCMC, is a feature-based estimation method (see Table 2 in Ref. [1]).*

– *Observation process distribution: “Possible difficulty of fitting negative binomial dispersion parameter”. As I wrote in my first review, in my experience this is typically not true and this parameter can be estimated very well. Admittedly, this parameter is more important to consider in deterministic model variants, where it allows capturing noise in the data.*

– *Transition rates between compartments: in all rigor, it is the duration spent in a given state that follows an Exponential or an Erlang distribution (not the rates).*

– *Choosing which parameters to fit and which to fix: to the best of my knowledge, I don’t know an occurrence of a model in which all the parameters were fitted from the data.*

- Thank you for all of these suggestions and details. We have moved the table to the supplement as suggested and have made each of these requested adjustments.

Figure 2: as the parameters indicate the rates of transition between compartments, in all rigor β_i should be replaced by the force of infection.

-- We have replaced β_t with $\beta_t * I_{\text{Eff}}/N$ and define the I_{Eff}/N term in the figure caption.

The authors use the adjective “hypoexponential” to describe the distribution of the onset-to-death time. Although I understand what is meant here, this adjective is not very common and could be replaced by something more explicit (like the actual name of the distribution).

- First, we have updated the text to now correctly state that times within states are geometrically distributed and not exponentially distributed given that we have a discrete time model (re: Reviewer 3's comment). Second, given that the sum of geometric distributions with different probabilities is not a named distribution with a closed form, we have dropped all names associated with the onset-to-death distribution. We state this on **lines 134-136**.

Figure S6: I think it would be more effective to use a scatterplot matrix to represent the estimates. Also, please provide the definition of relative log-likelihood in the legend. (If that simply represents the difference with the maximum log-likelihood, “scaled log-likelihood” is more commonly used to refer to that quantity.)

- First, given Reviewer 1's request for profiles, we have updated Figure S6 (**now Figure S7**) to show: the maximum log likelihood at each unique value of each Sobol-sequenced parameter; point clouds for the computed reproduction numbers. Second, we have added scatter plot matrices for the three dates (April 1, May 13, June 24) that we highlight in other supplemental figures. These scatterplot matrices are now **Figures S8-S10**. Finally, we have replaced the term "relative log-likelihood" with “scaled log-likelihood” in the figure and defined it in the caption. Thank you for pointing out this omission.

RESPONSE TO REVIEWER 3

1. In table 1, the authors summarize possible modeling options and their tradeoffs. These modeling decisions are discussed throughout the methods section, but the table itself is rarely discussed. I wonder if the authors can try to highlight the table more explicitly (perhaps in the discussion) so that other modelers can refer to the table in making modeling decisions.

- We have now moved Table 1 to the supplement (as suggested by Reviewer 2), which should help make it a bit less focal and thus indirectly help the lack of references to it. That being said, we have also added one reference to the table in the Discussion: Early modeling and lessons for future epidemic control (**line 476**).

2. The discussion section primarily focuses on the limitations and values of early models. But I wonder if there is anything that the early models could have done better at the beginning of the current pandemic (even with limited information). For example, is there anything that the authors would have done differently in hindsight? Or were there assumptions that other early models made but turned out to be wrong or inaccurate? It seems like it would be important to include a brief discussion on ways to improve early models---and such discussion will be informative for early models of future outbreaks.

- We have added one additional paragraph reflecting on our decisions (geometrically distributed transition periods, Poisson observation process, and no uncertainty in the estimated parameter values) (**lines 475-483**) and one additional paragraph giving our suggestions for future early modeling (in brief, more intentional collaborations and workflows to minimize duplication of effort, importance of collaboration between disease modelers and behavioral scientists, guidelines for proper incorporation of mobility data, greater engagement between policymakers and scientists, particularly to clarify types and timings of interventions being considered) (**lines 517-528**).

3. The added supplementary figures illustrate convergence to some degree. For example, R_0 estimates (Figure S6) show a clear likelihood curvature, demonstrating that parameters are well identified. Parameters that are sampled, rather than estimated, give flat likelihood curves, meaning that the data provide little information about those parameters, suggesting that they are difficult to estimate (as explained by the authors in the manuscript; and this is OK). However, I find parameter estimates for E_0 a bit weird. It seems that log-likelihood values increase as E_0 increases, but parameter estimates rarely go over $E_0 > 6$. I would expect log-likelihood values to peak eventually and decrease as E_0 continues to increase. I wonder if this has to do with the fact that E_0 was sampled between 0 and 6, but the *mif2* algorithm was run using either an insufficient number of iterations or a small standard deviation (not allowing the parameters to deviate much from their starting values). This might be a relatively minor issue given that E_0 is not a focal parameter but it is still concerning as it suggests that fits may not have converged. For example, R_0 may look like it converged at a given E_0 value (e.g., at some likelihood slice), but if E_0 has not converged yet, then other parameter estimates can be suboptimal. Also, please clarify how relative log-likelihoods are defined.

- We agree that the increasing log-likelihood with increasing E_0 (previously Figure S6, **now Figure S2** updated to show parameter profiles based on Reviewer 1's suggestion) is odd and was incompletely addressed in the previous manuscript version. We have now run diagnostics to assess both the potential for better convergence with different *mif2* settings, as well as the potential impacts a wider range of E_0 values could have on the other parameters by testing the effect of: 1) increasing the range of starting values for *mif2* initialization (from 0-6, to 0-30); 2) increasing the random walk standard deviation (*rw.sd*) for E_0 from 0.02 to 0.05 or 0.1 (we chose to increase the random walk standard deviation rather than increase the *mif2* iterations given the flatness of the last portion of the E_0 *mif2* traces (previously Figure S5, **now Figure S6**)). We run these for the central fit date (May 13, 2020) and present the findings of these diagnostics in the new **Figure S3**. As expected, we find that increasing the *mif2* *rw.sd* and the range of starting values does increase the range of final E_0 values (**Figure S3, A**). Further, we find that log-likelihood does continue to increase with increasing E_0 , but at a decreasing rate (see the likelihood profiles in **Figure S2** and **Figure S3, A, bottom right panel**). However, the identified larger E_0 values that are associated with higher likelihoods are all with later epidemic start dates. Thus, the increased flexibility in E_0 (larger *rw.sd* and wider range of starting values) largely acts to allow the model to

identify late epidemic starts to be consistent with the model and the observed data. In fact, the most likely parameter set identified consisted of $E_0 > 500$ with an epidemic start in late February. Given our knowledge of the epidemic in Santa Clara (including a woman who died on February 6 with no recent international travel who was later identified as infected with SARS-CoV-2 <https://www.nytimes.com/2020/04/22/us/santa-clara-county-coronavirus-death.html>) and the implausibility of so many exposed individuals showing up simultaneously, we believe that broadly these parameter sets with large E_0 and late epidemic start are far less biologically plausible.

- Further, we find that even with the changes to *mif2* settings, there is little agreement between replicate *mif2* runs for the same Sobol-sequenced parameter set (**Figure S3, B, different colors**), suggesting we have little ability to infer this parameter, likely due to: the rapid decay in information on initial value parameters, the long time period between start of the epidemic and first observation, and the under-reporting of deaths that likely occurred in the first weeks or months of the pandemic.

- Finally, we examined the distribution of the other fitted parameters and reproduction numbers for the original *mif2* settings and new *mif2* settings tested (**Figure S3, C**). In particular, in **Figure S3, C**, we show the parameter sets within 2 log-likelihood units of the best parameter set for each combination of *mif2* settings (colors) to approximate what parameter sets would have been included if those *mif2* settings were used for model fitting. We find that while the *mif2* settings can have a large effect on E_0 , they have little effect on the distribution of the other fitted parameters or the calculated reproduction numbers. As a result, we retained the original *mif2* settings used in the manuscript for all parameter CI and forecasting intervals. We summarize these findings in the Results (**lines 323 - 332**) and note this as a shortcoming of our model in the Discussion (**lines 497-500**).

-- We have also replaced "relative log-likelihoods" with the more common "scaled log-likelihoods" as suggested by Reviewer 2 and defined it in the caption (**now Figure S7**).

L125: "Early models must be built and calibrated to data rapidly with highly incomplete and uncertain data" redundant phrasing?

- Phrasing updated to reduce redundancy

L126: "We divided the population into states with respect to SARS-CoV-2" do you mean "with respect to SARS-CoV-2 infection"?

- "infection" added

L131: "We assumed that transitions between states were simulated as binomial or multinomial processes, which treat periods within each state as being exponentially distributed" Does this not correspond to geometric distribution, rather than exponential, given that it's a discrete-time model?

- Thank you, we have corrected this.

L151: "We include σ_{WFH} and the 152 work-from-home start date as two of the sampled parameters (see Table 3) but allow σ_{SIP} to be estimated by the model (see "Fitting the Model" below)." It should be justified why σ_{WFH} was sampled, instead of being estimated.

- We now note (on what is now **line 154-156**) that due to concerns that σ_{WFH} is not identifiable (separable from β_0) because (like E_0 , which we also had significant trouble with) it operates only very early in time and alters the transmission only prior to the first observed death in Santa Clara county.

L185: "This median is between ..." I understand this sentence, but it is awkward and confusing (especially the "and used in" part). Please rewrite.

- Rewritten

L205: "scaling by the estimated β_0 by σ_{SIP} and the median proportion of the population" => "scaling the estimated β_0 by σ_{SIP} and ..."?

- Sentence revised to improve clarity

L206: Why 200 simulations here instead of 1200?

- We used the median proportion susceptible remaining across 200 simulations *for each of the 1200 parameter sets* to scale R_{eff} . To clarify this we have added the phrase "for each of the 1200 parameter sets" and specify "200 simulated epidemics for the given parameter set".

L220--223: Shouldn't the number of epidemic forecasts correspond to 625 times the number of parameter sets?

- Yes. We see that our sentence was unclear. We have now added a clause at the end of the sentence stating we use 625 per parameter set, the number of which are variable by fit date.

L229: "but narrow because we ignored uncertainty in the parameters listed in Table 1, and thus should be interpreted with caution" It is also narrow because uncertainty in estimated parameters for each fit is not taken into account (though, in practice, these uncertainties might be all washed away by simulating across multiple parameter sets and may not matter too much).

- We have added a reference to uncertainty in estimated parameters for each fit being ignored in this sentence. We have also noted in the Discussion section about our sub-optimal choices (**lines 475-483**) that if we were to do this again we would use importance sampling to incorporate uncertainty in fitted parameters.

L230: This should be table 2.

- Updated to refer to the correct table

L234--239: Please divide this sentence up---it is difficult to understand. It is also not at all clear what the scoring is trying to measure. For example, why do you only consider values that are exactly equal to the observed values? Why does F suddenly appear? Why does the first summation not have any indices?

- We have entirely rewritten (and expanded) the section about the quadratic score to improve clarity (**lines 245-268**). As stated in Czado et al. 2009 [58] and Gneiting and Raferty 2007 [57], a quadratic score (one example of a strictly proper score) to quantify forecast accuracy is sensible for forecasts of count data as the quadratic score will increase with both the accuracy (described as calibration) and precision (described as sharpness) of the forecast.

L241: It's not clear which quantity is being compared with cases at this point in the text. Please clarify.

- Sentence clarified.

L261: "Counterfactuals" This section needs to be explained more clearly and requires a better preface. It is confusing what the purpose of this section is when the authors suddenly talk about diverging counterfactual scenarios (even for a reviewer who has already read through the paper once). It's also not exactly clear what the third counterfactual is trying to achieve (presumably to show that intervention introduced later will require additional measures to match the effectiveness of the intervention introduced earlier?).

- We have added two introductory sentences (**lines 290-295**) to clarify the purpose of the counterfactuals (in brief, to contextualize for local policymakers and the public the impact of early decisions relative to other possible decisions that could have been made---to quantify the impact of early decisions and to highlight which alternative decisions could have made the early epidemic outcomes better or worse).

L294--308: I wonder if it's possible for the authors to add RE values in Figure 4 (maybe in the top left corner of each panel) instead of referring to Figure 3. It's difficult to go back and forth two figures to try to figure out the RE values that correspond to the scenarios presented in Figure 4, especially given that Figure 4 is presented at a 3 week interval, whereas Figure 3 is presented at a 1 week interval.

- We have added an additional thin panel between the trajectories and quadratic score panels.

L306--308: I don't see any underestimation... Where is this result presented?

- The blue shaded violin plots in **Figure 3** show predicted deaths lower than observed deaths. We have added this detail with the reference to **Figure 3** in this sentence

L309: In reality, there are presumably considerable lags between when people develop symptoms and when cases are reported (and there were probably under-reporting of symptomatic cases as well). So I'm surprised that the number of symptomatic infections fits well with reported cases. Any comments from the authors? This is very briefly mentioned in L395 ("lagged one week for reporting lag") but I think it should be explained earlier (in Model assessment section).

- We have added a sentence about the one week lagged cases in the second paragraph of "Model assessment" (**lines 267-268**) and generally expanded the description about case evaluation in this section (**lines 264-268**). We now also point out the difference in vertical axes in **Figure 5** in the Results for model-predicted new symptomatic infections and reported cases which de-emphasizes the relative magnitudes of the values in favor of a simple visual comparison of the curvature of the trajectories (**lines 355 - 356**).

L320: "We originally designed and fit our model" => "designed and fitted"?

- Fixed

L407: "This pattern occurred throughout the U.S. during the summer resurgence" any citations?

- Citation added that found lower mortality rates in many US states during the second wave.

Table 1: "More states increases the number of parameters and will be harder to fit" increase => increase?

- Fixed

Table 1: "Exponentially distributed rates vs Erlang distributed rates" This sounds like rates are distributed. Please clarify.

- Fixed

Figure 1: "SafeGraph" is never mentioned in the text. Maybe remove unnecessary details to avoid confusion?

- We prefer to keep the bottom panel in Figure 1 because we find that it helps provide a visual for the changing epidemiological environment that we discuss as a primary reason why the model begins to fail. Given that we want to keep quantitative data in the lower panel a citation is necessary. We use language in the caption to clarify that the data are not actually used in the model.

Figure S6: axis labels overlap

- Fixed (now **Figure S7**)